# Reciprocal expression of MADS-box genes and DNA methylation reconfiguration initiate bisexual cones in spruce

Yuan-Yuan Feng [ID] [1,2,3], Hong Du[1,2], Kai-Yuan Huang [ID] [1,2,3], Jin-Hua Ran [ID] [1,2,3,4 ✉] & Xiao-Quan Wang [ID] [1,2,3,4 ✉]

The naturally occurring bisexual cone of gymnosperms has long been considered a possible intermediate stage in the origin of flowers, but the mechanisms governing bisexual cone formation remain largely elusive. Here, we employed transcriptomic and DNA methylomic analyses, together with hormone measurement, to investigate the molecular mechanisms underlying bisexual cone development in the conifer *Picea crassifolia*. Our study reveals a "bisexual" expression profile in bisexual cones, especially in expression patterns of B-class, C-class and *LEAFY* genes, supporting the out of male model. *GGM7* could be essential for initiating bisexual cones. DNA methylation reconfiguration in bisexual cones affects the expression of key genes in cone development, including *PcDAL12*, *PcDAL10*, *PcNEEDLY*, and *PcHDG5*. Auxin likely plays an important role in the development of female structures of bisexual cones. This study unveils the potential mechanisms responsible for bisexual cone formation in conifers and may shed light on the evolution of bisexuality.

[1] State Key Laboratory of Plant Diversity and Specialty Crops, Institute of Botany, Chinese Academy of Sciences, Beijing 100093, China. [2] China National Botanical Garden, Beijing 100093, China. [3] University of Chinese Academy of Sciences, Beijing 100049, China. [4] These authors contributed equally: Jin-Hua Ran, Xiao-Quan Wang. ✉email: jinhua_ran@ibcas.ac.cn; xiaoq_wang@ibcas.ac.cn

The "sudden" origin and explosive diversification of angiosperms, which dominate terrestrial and many aquatic ecosystems worldwide, were associated with the emergence of several innovative characteristics. Among these, flowers are considered as the key reproductive organ innovation[1–3]. Since Darwin's era, the origin and evolution of flowers have been a mystery that has attracted much attention, leading to the proposal of a series of hypotheses. Based on the fact that all land plants but most angiosperms possess unisexual reproductive structures[3], together with the current phylogenetic framework and morphological evidence[4,5], bisexuality is considered likely the first step in the origin of flowers followed by compression of the floral axis[3,6,7], although male and female reproductive organs could be secondarily separated like in diecious angiosperms[3]. This is also supported by fossil evidence that the reproductive axes of *Archaefructus sinensis* lacked sepals and petals and produced stamens in pairs below conduplicate carpels[8]. Despite widespread acceptance of the euanthial theory[9] and reconstruction of the ancestral angiosperm flower using the largest dataset of floral traits ever assembled[10], controversies persist regarding the morphology of the ancestral angiosperm flower[11]. In particular, how flowers originated step by step and the molecular mechanisms governing their development remain unresolved[11–14]. Different theories and models, including the mostly male theory, the out of male model and out of female model, have been proposed to explain the evolution of "the first flower" (the origin of bisexuality and compression of the reproductive axis) based on the ABC model of angiosperm flower development and the structures of the male and female cones of gymnosperms[3,14–17]. However, an extreme morphological gap exists between extant gymnosperm cones and angiosperm flowers, which could not be filled by fossil records[3,18]. This makes it difficult to test the reliability of these models by comparing the reproductive structures of extant seed plants.

Recently, the molecular mechanisms of the initiation of bisexual cones (upper ovules, basal pollen sacs) in gymnosperms have attracted increasing attention[19,20]. Although exceedingly rare in nature[21,22], bisexual cones have been documented in different lineages of gymnosperms, particularly in conifers like *Agathis, Larix, Picea, Pinus, Phyllocladus,* and *Saxegothaea*[22–27], suggesting that the bisexual structure likely originated from the common ancestor of gymnosperms and angiosperms[23]. A rare occurrence of bisexual structure in gymnosperms may result primarily from natural selection to avoid inbreeding constraints, given the absence of an incompatibility system[23]. The euanthial theory supposed that flowers are uniaxial structures, with carpels and stamens homologous to gymnosperm macrosporophylls and microsporophylls, respectively[9]. This resemblance to ancestral perianth-less bisexual flowers is evident in morphology of bisexual cones in gymnosperms[17,28,29]. Consequently, bisexual cones have long been considered an intermediate state[29], and have been used to elucidate the transition to hermaphroditism during the origin of flowers[6,17]. Niu et al.[19] investigated the bisexual cones of *Pinus tabuliformis*, finding both male and female structures functional. According to their study, the transcriptomes of male structures are more similar to those of female cones, indicating a "mostly female" gene expression profile in bisexual cones. In addition, Feng et al.[20] revealed that hormonal regulation could be related to the sexual reversal of bisexual cones in *Pinus massoniana*. However, none of these studies conducted separate RNA-seq analysis on the female and male structures of bisexual cones, hindering comparative analysis of gene expression patterns between these structures. Moreover, in the study of Niu et al.[19], the sampling was not performed early enough to identify direct regulators and extensively explore the molecular mechanisms responsible for the initiation of bisexual cones. Therefore, an in-depth and comprehensive study on the differentiation of bisexual cones in gymnosperms is crucial for understanding the origin of bisexuality.

In addition to the MADS-box genes and hormones that play important roles in flower development, recent studies indicate that DNA methylation is also involved in the development of reproductive structures. Li et al.[30] found that the upregulated genes *AoMS1, AoLAP3, AoAMS,* and *AoLAP5*, with varied methylated CHH regions, might have been involved in sexual differentiation in *Asparagus officinalis*. Specifically, during the meiotic stage, *AoMS1* and *AoLAP3* showed hypomethylated CHH differentially methylated regions (DMRs) in male flowers, in contrast to female flowers. Additionally, at the meiotic stage of male flowers, *AoAMS* and *AoLAP5* exhibited hypermethylated CHH DMRs, distinguishing them from the premeiotic stage. Martin et al.[31] reported that DNA methylation changes in the promoter of *CmWIP1* resulted in the transition from male to female flowers in melon. Moreover, Akagi et al.[32] discovered that DNA methylation regulated the influence of *MeGI* (Japanese for female tree) and *OGI* (Japanese for male tree) on sex determination in *Diospyros kaki*, a hexaploid persimmon. Nevertheless, the study of DNA methylation in gymnosperms lags far behind that in angiosperms due to their large genomes and abundant repetitive sequences[33]. Previous studies have shown that global methylation levels of CG and CHG in gymnosperms are much higher than those in angiosperms[34–36], and Flores-Rentería et al.[21] suggested that genetic plasticity might contribute to the formation of complex sexual systems in *Pinus johannis*, because sexual inconstancy was only detected in some unisexual individuals. However, the role of DNA methylation in initiating bisexual cones of gymnosperms remains unexplored.

In this study, we conducted a comprehensive comparative analysis of transcriptomic, DNA methylomic and hormonal variation in different developmental stages of normal male and female cones and bisexual cones in *Picea crassifolia* (Qinghai spruce), a monecious species of Pinaceae. We aim to: (1) evaluate which hypothesis of origin of bisexuality is supported by the transcriptome and MADS-box gene expression patterns of bisexual cones; (2) reveal the role of DNA methylation in bisexual cone initiation and its influence on cone development-related genes; and (3) investigate hormonal influence on the development of bisexual cones.

## Results and discussion

**The out of male model is supported, and *GGM7* is essential for initiating bisexual cones.** Sporadically occurring bisexual cones, which may represent an intermediate state in the origin of flowers and the primitive form of hermaphroditic flowers[23,29], have been documented in different clades of gymnosperms. Previous and current studies indicate that the male structures of bisexual cones can produce functional pollens (Fig. 1a, b and Supplementary Table 1), while female structures have two different fates[23]. In case where female structures occupy a small proportion, the bisexual cones will dry and drop without seed production. In contrast, if the female structures occupy a large proportion, the bisexual cones can produce seeds, as reported in *Abies balsamea, Pinus densiflora* f. *multicaulis, P. johannis*, and *P. tabuliformis*[19,23,37,38]. This implies that bisexual cones may not necessarily be deleterious and nonfunctional. Despite some studies on bisexual cones, the underlying mechanisms remain largely unknown.

In this study, RNA-seq analysis of unisexual and bisexual cones in *P. crassifolia* revealed highly consistent expression profiles for the biological replicates of male cone (M), female cone (F), male and female structures of bisexual cone (BM and BF) at each stage

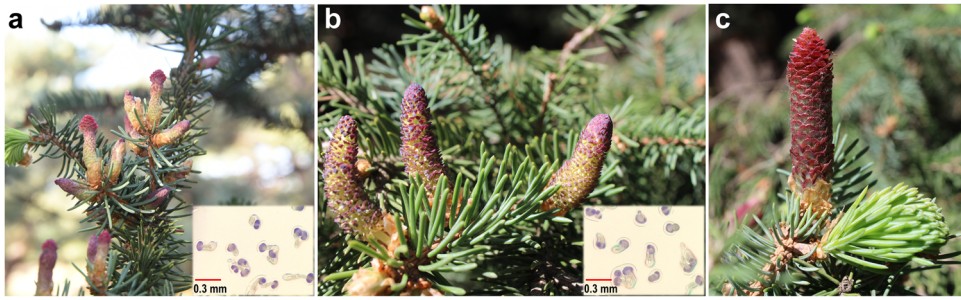

**Fig. 1 Phenotypes of unisexual and bisexual cones of *Picea crassifolia*. a** Bisexual cones. **b**, **c** Unisexual male and female cones. Pollen germination from male structures of bisexual cones and male cones is shown in the lower right corner of (**a**) and (**b**), respectively. **a**–**c** were recorded on April 13, 2019.

**Fig. 2 Samplings and expression characterization (RNA-seq) of unisexual and bisexual cones. a** Samplings of unisexual and bisexual cones. Stages 1–6 were collected on 19 March, 25 March, 1 April, 7 April, 13 April, and 18 April, 2019, respectively. M, male cone; F, female cone; B, bisexual cone. **b**, **c** Pearson correlation coefficients and principal component analysis of samples from the fourth and fifth developmental stages of unisexual and bisexual cones. **d** Number of DEGs between pairs sampled at the fourth developmental stage.

(Fig. 2b, c and Supplementary Fig. 1a, b, d, e). The transcriptome expression profiles of BF and BM are similar to those of F and M, respectively (Fig. 2b, c). Differential expression analysis revealed a gradual increase in differentially expressed genes (DEGs) from M2 to M5 relative to M1. In contrast, F3 relative to F1 had more DEGs compared to F2, F4, and F5 relative to F1, possibly due to rapid F3 growth (Supplementary Fig. 1c, f). In addition, among M4, F4, BM4, and BF4, F4 vs. BF4 exhibited the fewest DEGs (Fig. 2d). These results indicate a similarity between the transcriptome of BF and BM and their respective counterparts, F and M, demonstrating a "bisexual" transcriptome expression

pattern in bisexual cones. Niu et al.[19], however, investigated the expression profiles and regulatory mechanisms of bisexual cones in *P. tabuliformis* using RNA-seq and microarray analysis and found a transcriptomic similarity between male structures of bisexual cones and female cones, showing a "mostly female" gene expression profile in the bisexual cones. Using the *P. tabuliformis* genome as reference data, we reperformed clustering analyses, including principal component analysis, Pearson correlation coefficient and hierarchical clustering analysis of transcriptome expression profiles, and hierarchical clustering of MADS-box genes expression profiles. These analyses were conducted on the

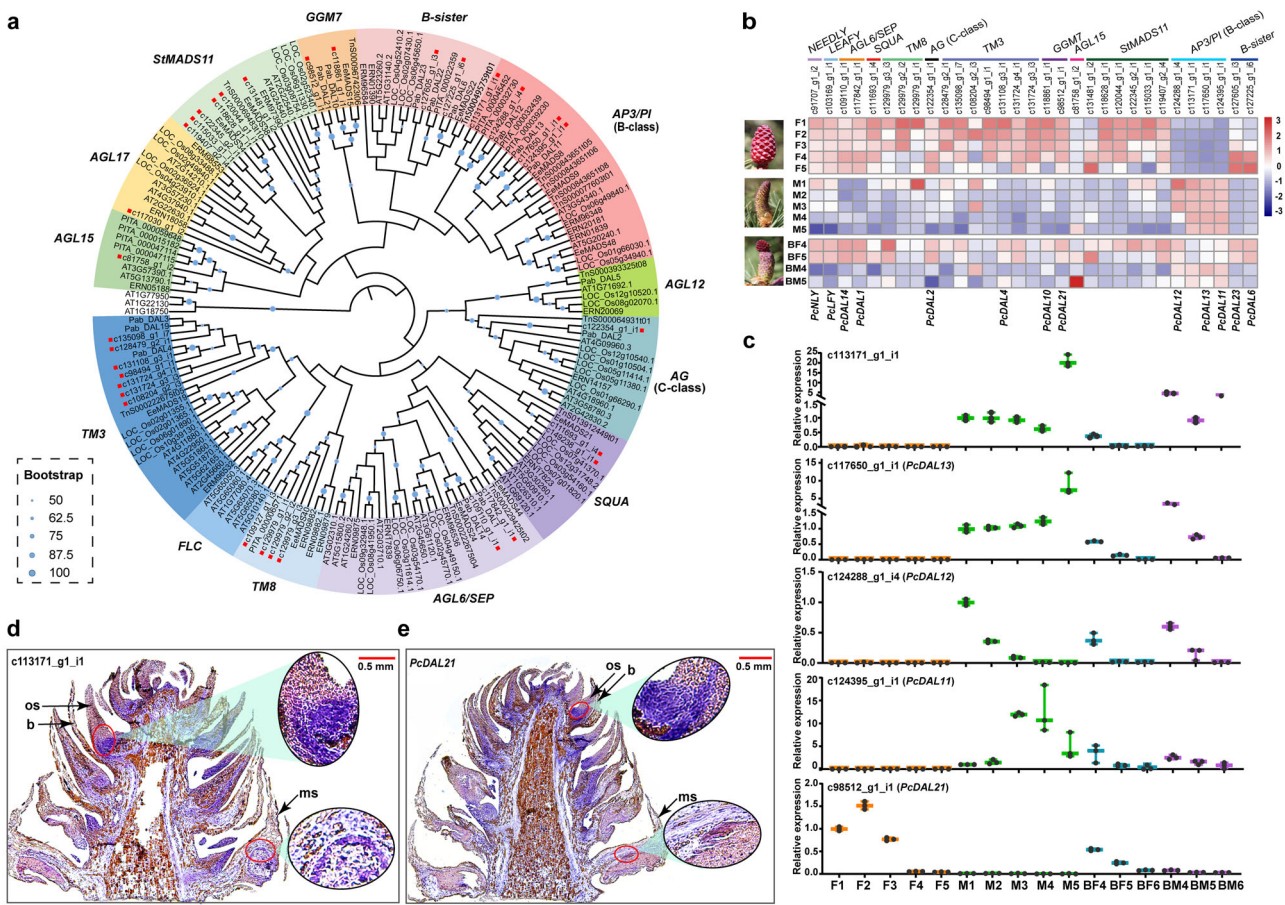

**Fig. 3 Phylogenetic relationships and expression profiles of MADS-box genes. a** ML tree of MIKCc MADS-box genes. **b** Expression heatmap of MADS-box genes, *PcLFY* and *PcNLY*. Female and male cone photos were taken on 13 April, 2019, while the bisexual cone photo was taken on 10 April, 2019. **c** qRT–PCR analysis of B-class and *PcDAL21* genes ($n = 3$ biological independent duplications). The horizontal bars, denoting the maximum, median, and minimum values, are positioned at the top, middle, and bottom, respectively. **d**, **e** In situ localization of c113171_g1_i1 and *PcDAL21* in a bisexual cone collected on 7 April, 2019. os, ovuliferous scale; b, bract; ms, microsporophyll.

expression data from all samples reported in Niu et al.[19] (data provided by Shihui Niu). Our reanalysis confirmed concordance of the transcriptome and MADS-box gene expression profiles between the male structures of bisexual cones and male cones (Supplementary Fig. 2), as found in *P. crassifolia*. Discrepancies in clustering results from Niu et al.[19] may stem from methodological deviations, given that they used only 3989 DEGs for cluster analysis whereas our reanalysis incorporated over 20,000 genes.

In addition, we identified 32 candidate MIKCc MADS-box genes from the Reference Sequence (RefSeq) database that were divided into 11 clades, including *AP3/PI* (B-class) (4), *B-sister* (2), *AG* (C-class) (1), *SQUA* (2), *AGL6/SEP* (2), *TM8* (4), *TM3* (7), *AGL15* (1), *AGL17* (1), *StMADS11* (6), and *GGM7* (2) (Fig. 3a and Supplementary Data 1). Since the ABC model established in *Arabidopsis* is also applicable to gymnosperms, where B- and C-class genes govern male cone development, while C-class genes control female cone formation[39], Theißen et al.[17] proposed the out of male and out of female models based on B-class gene expression changes to explain hermaphrodite formation. The former suggested that reduced B-class gene expression in the upper part of male cone led to ectopic ovule development, while the latter assumed that ectopic expression of B-class genes at the base of female cone resulted in the ectopic development of male reproductive units. In our study, most MADS-box genes in BF showed expression patterns resembling those in F, except for B-class genes. Even among B-class genes, *PcDAL12* (c124288_g1_i4) in BF4 displayed slightly higher expression than

in M4, whereas other BF4 members exhibited expression levels higher than in F but lower than in M4 and BM4 (Fig. 3b, c). These findings were corroborated by in situ hybridization, where the specific signals of B-class genes appeared in the base of the female structure and the tapetum of the male structure in bisexual cones (Fig. 3d and Supplementary Fig. 3). In addition, observations over 6 years showed that almost all bisexual cones grew in the same locations as male cones (Fig. 1a), consistent with the findings of Caron and Powell[22] and Flores-Renteria et al.[23]. These observations, including transcriptome expression profiles and B-class gene expression patterns, support the out of male model.

In 2006, Baum and Hileman[6] supplemented the out of male model by proposing that increased *LEAFY* (*LFY*) gene expression along the reproductive axis results in differential regulation of B- and C-class genes during flower development. Consequently, B-class genes exhibit high expression at the base of the reproductive axis and low expression at the top, whereas C-class genes show the opposite pattern, leading to the formation of diverse floral quartet-like complexes at the base and top of the reproductive axis[6]. When validated through quantitative real-time PCR (qRT–PCR), our study demonstrated elevated expression of *PcLFY* and *PcAG* (C-class) in BF, while B-class genes exhibited heightened expression in BM during the initiation stage in *P. crassifolia* (Fig. 3b, c and Supplementary Fig. 4a), supporting the hypothesis of Baum and Hileman[6]. In addition, the *PcLFY* and *PcNEEDLY* (*PcNLY*) genes were expressed in both normal

male and female reproductive structures (Fig. 3b and Supplementary Fig. 4a, b), with higher levels in females. This challenges the mostly male theory, which hypothesized that *LFY* and *NLY* genes determined male and female reproductive structures, respectively, in the most recent common ancestor of seed plants[5].

Recent studies have proposed that the interaction patterns of proteins encoded by MADS-box genes in gymnosperms are similar to the floral quartet model observed in angiosperms. This similarity suggests that the interaction of MADS-box proteins was already established in the most recent common ancestor of extant seed plants[16]. However, gymnosperms, as the sister group of angiosperms that diverged approximately 300 million years ago[4,40], exhibit distinctions in the MADS-box genes governing flower development. *G. gnemon MADS7* (*GGM7*) was firstly cloned and sequenced from *Gnetum gnemon*[41], with possible corresponding orthologous genes found in ferns and bryophytes[42,43]. However, it lacks a counterpart in angiosperms[44]. In this study, we identified two *GGM7* genes, *PcDAL21* (c98512_g1_i1) and *PcDAL10* (c118861_g1_i1), homologous to *PaDAL21* and *PaDAL10* of *Picea abies*, respectively. *PcDAL21* exhibited negligible expression in male cones but showed substantial expression in early-stage female structures, with higher expression levels observed in BF4 than in BM4, presenting a contrasting expression pattern to that of B-class genes (Fig. 3b, c). This expression pattern was confirmed by in situ localization, where specific *PcDAL21* signals were localized predominantly at the base of the female structure within the bisexual cone, while being relatively weak in the microsporophyll (Fig. 3e). The specific expression in female cones of *DAL21* has also been observed in *P. abies*[44] and *Cunninghamia lanceolata*[45], indicating a correlation between *DAL21* expression initiation and the onset of ovuliferous scale primordia. Given the elevated expression at the base and decreased expression at the apex of B-class genes in bisexual cones, we infer that *PcDAL21* may function in female cones, establishing female identity and potentially antagonizing B-class genes. However, further experiments are needed to confirm *DAL21*'s role in forming floral quartet-like complexes in female cones of *P. crassifolia* or even gymnosperms. The expression of *PcDAL10*, another member of *GGM7*, was higher in bisexual cones than in male and female cones at corresponding developmental stages (Fig. 3b and Supplementary Fig. 4c). Previous studies revealed specific *PaDAL10*'s expression in reproductive structures of *P. abies*[44]. Transgenic *Arabidopsis* plants expressing *PaDAL10* exhibited notable morphological changes in sepals, petals and stamens, suggesting its interaction with B- and C-class genes[46]. This hypothesis was validated through yeast two-hybrid assays involving *PtDAL10*, an orthologue of *PaDAL10* in *P. tabuliformis*, which exhibited widespread interactions with other MADS-box genes, including B-class, C-class, *SEP/AGL6*, and others[36]. Therefore, both ectopic expression of *PcDAL21* and increased expression of *PcDAL10* might be essential for the initiation of bisexual cones.

Additionally, *SEP* genes (E-class), which act as mediators in the formation of male- and female-specifying complexes in angiosperms[47], have not been identified outside the angiosperm lineage[48]. While some gymnosperm genes are phylogenetically close to the *AGL6* subfamily, a clade closely related to E-class genes, it remains a matter of debate whether these genes are true orthologs of angiosperm *AGL6* clade or if they are instead sister to the *AGL6/SEP* clade[14,49]. Notably, a study has shown that in *Gnetum gnemon*, B- and C-class proteins can directly interact without *SEP* or *AGL6* genes as mediators[39]. Besides that, studies have shown that *NLY* can recognize sequences containing a *LFY* binding motif, inducing flower formation and complementing the *lfy* mutant when expressed in *Arabidopsis thaliana* or *Nicotiana*

*tabacum*[50–52]. In the initial stage of bisexual cones in *P. crassifolia*, *PcNLY* expression was higher in the female structures compared to normal male and female cones (Fig. 3b and Supplementary Fig. 4b). Consequently, the absence of *GGM7* and *NLY* genes, and retention (or new functionalization) of E-class genes in angiosperms may lead to alterations in the interaction modes among MADS-box genes[5,39,42,43,53]. These changes could account for the distinct composition of complexes responsible for specifying male and female organ identities between gymnosperms and angiosperms, thus contributing to the formation of key floral traits in angiosperms.

**DNA methylation reconfiguration may contribute to the initiation of bisexual cones.** To reveal the role of DNA methylation in bisexual cone initiation and its influence on cone development-related genes, we conducted a comparative analysis of gene-body methylation (gbM) in male and female cones (M4 and F4), as well as the male and female structures of bisexual cones (BM4 and BF4) in *P. crassifolia*. All four tissues exhibited bisulfite conversion rates exceeding 97.7% (Supplementary Table 2), confirming the high quality of our sequencing data. The methylome data from three biological replicates for each tissue were pooled for subsequent analysis, given the strong correlation between the biological replicates (Supplementary Fig. 5). In general, BF4 exhibited the highest global CG and CHG methylation levels compared to the other three tissues, along with the lowest percentage of CG and CHG body-methylated genes. Nonetheless, there was no significant difference in the number of methylated sites across four tissues (Fig. 4a–c and Supplementary Fig. 6a–c). Notably, the percentages of CG and CHG body-methylated genes were highest in BM4 (Fig. 4c and Supplementary Fig. 6c). Through the analysis of DNA methylation site density in body-methylated genes, we found that genes with high CG and CHG methylation site density exhibited the highest occurrence in BF4 and the lowest in BM4 (Fig. 4d). These results indicate an altered global methylation pattern in bisexual cones compared to normal male and female cones. BF4 showed slightly increased global CG and CHG methylation levels, with concentrated methylated CG and CHG sites in specific unigenes, while BM4 exhibited the opposite trends, suggesting distinct methylation strategies between the two. In addition, we observed a high overlap of body-methylated genes across various tissues, with BF4-specific body-methylated genes being the least. Significant differences in CG and CHG methylation levels were noted in body-methylated genes in M4 vs. F4 and F4 vs. BF4 (Fig. 4e, f and Supplementary Data 2–5). Furthermore, there were more differentially methylated cytosines (DMCs) and DMRs in M4 vs. BM4 and F4 vs. BF4, with greater DNA methylation level differences of DMCs in M4 vs. BM4 and F4 vs. BF4 compared to M4 vs. F4 and BM4 vs. BF4 (Fig. 5a, b). These findings suggest a CG and CHG DNA methylation reconfiguration in bisexual cones, signifying genome-wide alterations in DNA methylation patterns without massive disappearance of DNA methylation[54]. Previous studies indicated prevalent DNA methylation alterations in sex determination across plants and animals, including sex-specific methylation signals identified in the genome of *Populus balsamifera*[55], and significant global methylation differences observed between the gonads of male and female individuals in *Crassostrea gigas*[56].

To further investigate the effect of DNA methylation on bisexual cone development, we identified genes with differential methylation and expression in M4 vs. F4 and BM4 vs. BF4 (Fig. 6a and Supplementary Data 6, 7). Overall, there was no significant correlation between methylation differences and expression differences. Nevertheless, a specific gene

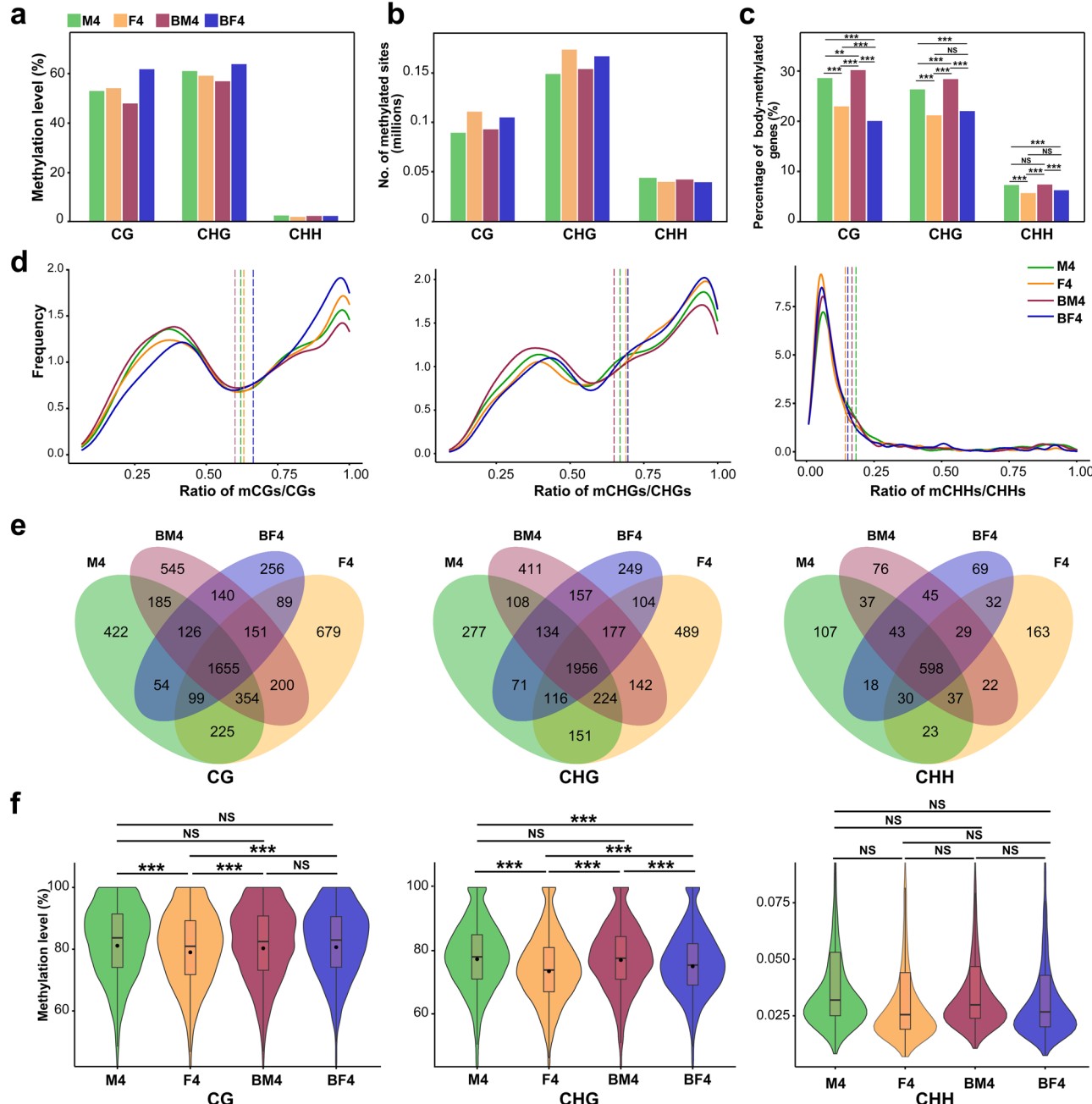

**Fig. 4 DNA methylation characterization of M4, F4, BM4 and BF4. a** DNA methylation levels. **b**, **c** Number of methylated cytosine sites and percentage of body-methylated genes. Statistical analysis was conducted using the chi-square test. **, *P* value < 0.01; ***, *P* value < 0.001; NS, *P* value > 0.05. **d** Frequency distribution of mC densities in body-methylated genes. The dashed lines represent mean values. **e** Venn diagrams of body-methylated genes. **f** Violin- and box-plots of the methylation level of body-methylated genes. Boxplot with lines representing median, 25th and 75th percentiles, with a central dot indicating the mean value. Statistically significant differences were analyzed using the two-tailed Student's *t* test. ***, *P* value < 0.001; NS, *P* value > 0.05. The methylation data were obtained by combining data from three biological replicates.

(c126976_g1_i5) displayed significant differences in both methylation and expression levels in M4 vs. F4 and BM4 vs. BF4. This gene, classified within the HD-ZIP IV subfamily (Supplementary Data 8) and featuring homeobox and START domains, was denoted as *PcHDG5*. Its nomenclature is derived from its close phylogenetic relationship with the *AtHDG5* and *AtHDG4* genes of *Arabidopsis thaliana* (Supplementary Fig. 7a and Supplementary Data 9). Kamata et al.[57] discovered that the double mutant *pdf2-1 hdg5-1* of *A. thaliana* produces flowers exhibiting sepaloid petals and carpelloid stamens, accompanied by a significant reduction in *AP3* gene expression. Therefore, we inferred that

*PcHDG5* may function as an upstream regulator of *PcAP3/PI*. We implemented a transient expression system in tobacco to assess the hypothesis. Quantitative analysis of dual-LUC transient expression assays showed a significantly higher LUC/REN ratio with cotransformation of PcHDG5-GFP and the reporter with the upstream region of *PcDAL13* (c117650_g1_i1) compared to the control (Fig. 6b and Supplementary Fig. 7b, c). Additionally, binding sites for *AtHDG5* were found with high confidence in the promoter region of *PcDAL13*, as confirmed by the JASPAR database (9th version, http://jaspar.genereg.net/)[58] (Supplementary Table 3). Therefore, *PcHDG5* might be the

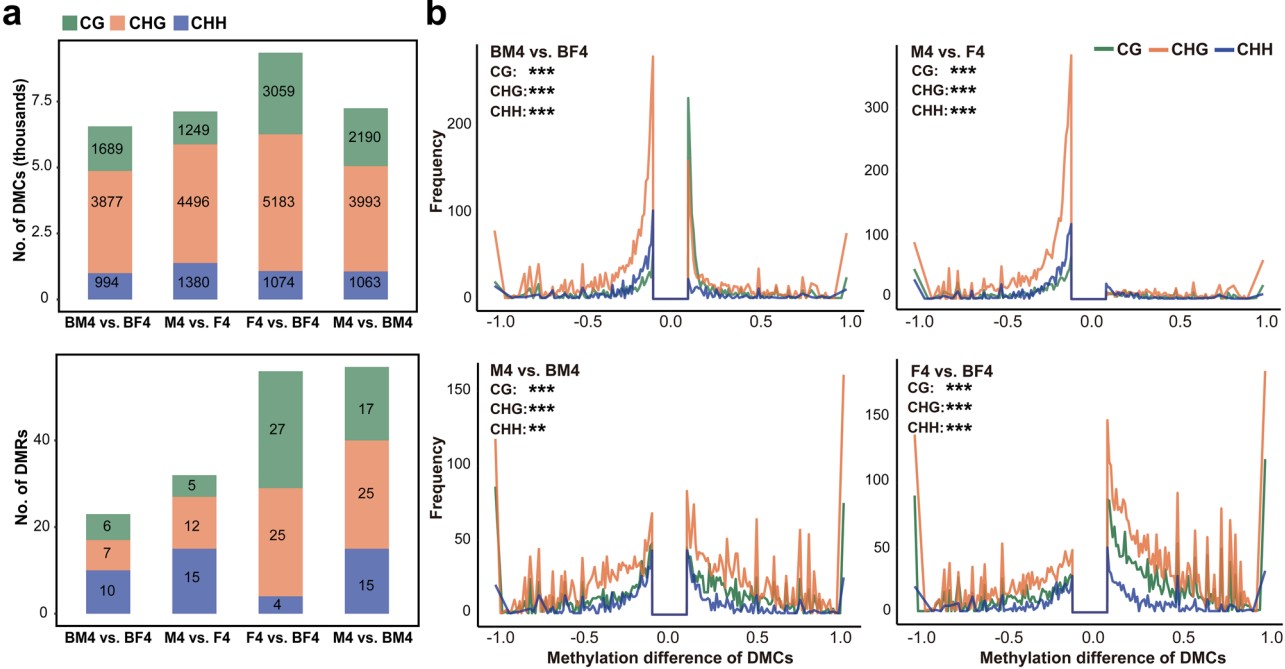

**Fig. 5 Gene-body DNA methylation differences among different tissues. a** Number of DMCs and DMRs. **b** Frequency distribution of differential methylation levels of DMCs. Statistically significant differences were analyzed using the two tailed Wilcoxon test. **, *P* value < 0.01; ***, *P* value < 0.001. DMCs, differentially methylated cytosines; DMRs, differentially methylated regions.

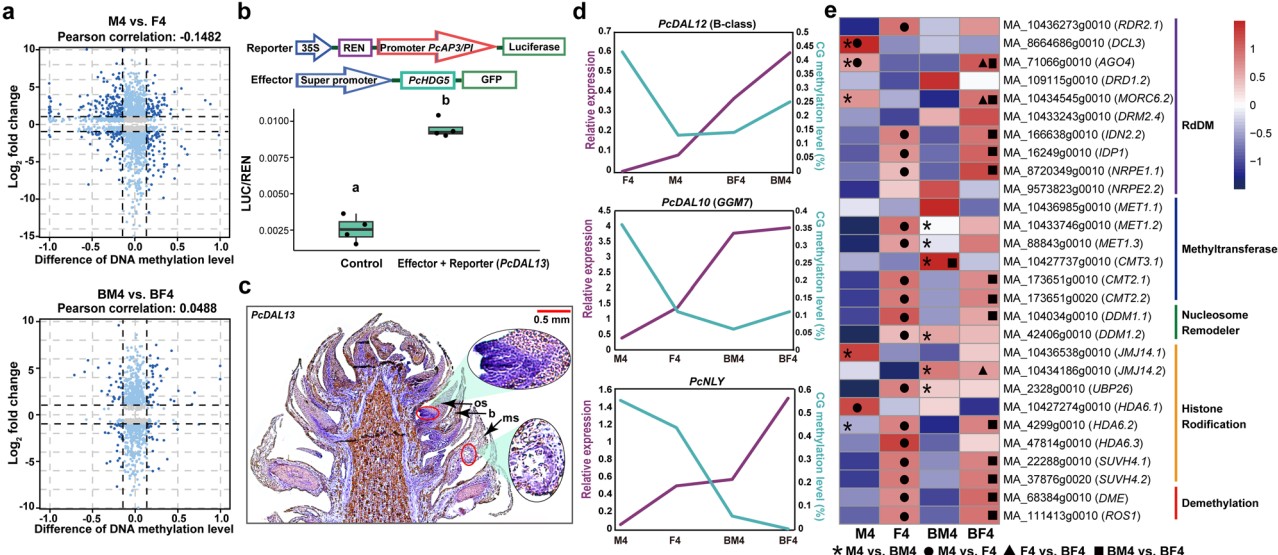

**Fig. 6 Combined analysis of the DNA methylome and transcriptome. a** Nine-quadrant diagram of the difference in DNA methylation and expression. **b**. Schematic of the effector and reporter and quantitative analysis of dual-LUC transient expression assays of *PcDAL13* promoter activity. Box plot lines represent, from bottom to top, 25th percentile, median and 75th percentile of the data dispersion. Statistical analysis was conducted using the two tailed Student's *t*-test (*n* = 4 biological independent duplications). Different letters denote significant differences at *P* value < 0.05. **c** In situ localization of *PcDAL13* in a bisexual cone collected on 7 April, 2019. **d** Line charts of CG promoter methylation level and relative expression of *PcDAL12*, *PcDAL10* and *PcNLY*. **e** Expression heatmap of genes involved in the methylation process. Statistical analysis was conducted using the two-tailed Student's *t*-test. Asterisks, dots, triangles, and rectangles represent genes with significantly upregulated expression (*P* value < 0.05) in M4 vs. BM4, M4 vs. F4, F4 vs. BF4, and BM4 vs. BF4, respectively.

upstream transcription factor, binding to the core-binding motif to activate *PcDAL13* (*AP3/PI*) transcription. *PcDAL13* is an orthologue of *PaDAL13* (*P. abies*) and *PtDAL13* (*P. tabuliformis*). *PtDAL13* has been demonstrated in yeast two-hybrid assays to be part of the regulatory network for male determination[36], and in situ hybridization showed that *PaDAL13* expression is related to microsporophyll development[59]. In the present study,

specific *PcDAL13* expression signals appeared in the base of the macrosporophyll and tapetum of the microsporophyll (Fig. 6c). Additionally, *PcDAL13* expression gradually decreased in the bisexual cones, mirroring the trend observed in *PcHDG5*. However, in the female cones, *PcHDG5* was highly expressed, whereas *PcDAL13* was not (Fig. 3c and Supplementary Fig. 4d). This difference could be attributed to the regulation of *PcDAL13*

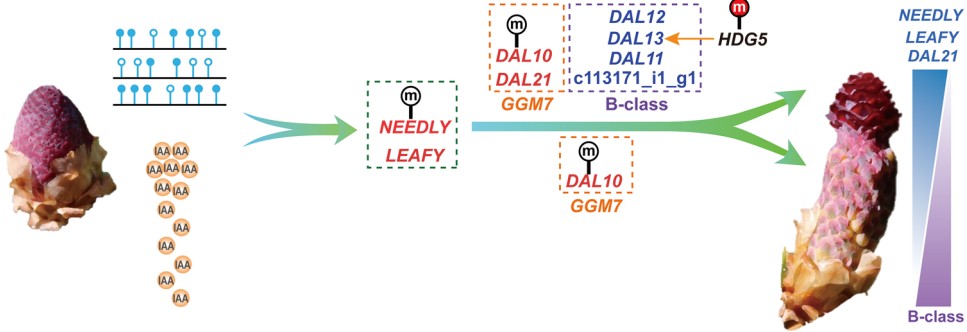

**Fig. 7 A predictive model of the molecular mechanism of bisexual cone initiation.** The graphical representation utilizes horizontal lines with solid and hollow dots to represent DNA methylation reconfiguration, while the orange circles labeled with IAA depict the concentration gradient of IAA. Upregulated genes are highlighted in red, while downregulated genes in blue. Promoter hypomethylation is denoted by "m" on a white background, whereas gene-body hypermethylation is indicated on a red background. Positive regulatory relationships are illustrated by orange arrows. The expression gradient of *DAL21*, *LFY*, and *NLY* in bisexual cones is represented by a gradual blue triangle, while the gradual light purple triangle shows the expression gradient of B-class genes in bisexual cones. The male cone was taken on 1 April, 2019, and the bisexual cone was photoed on 10 April, 2019.

by other genes like *LFY* and other MADS-box genes[36,50], or posttranscriptional control of *PcHDG5*[60,61], suggesting the complexity of the regulatory network governing gymnosperm cone development. We also examined DNA methylation patterns in the promoter regions of some important genes, such as MADS-box genes, *PcLFY*, *PcNLY* and *PcHDG5*. Although incomplete reference sequences resulted in missing or truncated gene promoters, we successfully obtained promoter sequences from 27 of these genes through PCR. Most of the genes exhibited no significant changes in methylation levels within their promoter regions (Supplementary Fig. 8). Nevertheless, CG methylation differences were found in the promoters of *PcDAL12* and *PcDAL10* (Fig. 6d and Supplementary Fig. 8a). *PcDAL12* displayed high CG methylation in its promoter with minimal expression in F4 (Fig. 6d), while *PcDAL10* showed the highest CG methylation level in the promoter and the lowest expression in M4 (Fig. 6d). *PcDAL12* is orthologous to *PaDAL12*, which may be involved in establishing male identity[44,59]. In addition, *PcNLY* had higher CG methylation levels in the promoter region and lower expression levels in normal cones (Fig. 6d), but CG methylation levels were not significantly different between F4 and BF4. These findings suggest potential regulation of DNA methylation on the crucial genes *PcDAL12*, *PcDAL10*, *PcNLY*, and *PcHDG5* to initiate bisexual cones.

The dynamic regulation of DNA methylation might be associated with changes in expression levels of genes related to de novo methylation, maintenance methylation, and demethylation. Particularly, the RdDM pathway, initiating de novo methylation, affects methylation broadly across all sequence contexts[62–64]. Comparing methylation-related gene expression across tissues revealed similarities in de novo methylation, maintenance methylation and demethylation genes between BF4 and F4. The majority of DNA methylation-related genes showed higher expression in BF4 than BM4, implying methylation process activation in BF4. Most genes involved in the RdDM pathway expressed at higher levels in BF4 than in F4, with significant differences in *AGO4* and *MORC6*[65,66], suggesting active RdDM pathway in BF4, which resulted in elevated CG and CHG methylation levels (Fig. 6e and Supplementary Fig. 9). In contrast, the RdDM pathway weakened in BM4 compared to M4, possibly reducing CG and CHG methylation, although upregulation of *MET1*[67] and *CMT3*[68] might enhance the ability to maintain these methylations. Further investigations are needed to understand the reasons behind the alterations in expression levels of these methylation-related genes.

**Auxin could enhance the femaleness of *Picea crassifolia*.** Interestingly, genes related to auxin signal transduction, transport and response were highly expressed in BF compared to BM during bisexual cone initiation. Auxin-related pathways were enriched in upregulated genes in F4 vs. BF4, M4 vs. BF4 and BM4 vs. F4, and in downregulated genes in M4 vs. BM4 and M4 vs. F4 (Supplementary Fig. 10a–g). Correspondingly, hormone analysis indicated significantly higher auxin content in BF than in BM during bisexual cone initiation (Supplementary Fig. 10h), emphasizing auxin's role in promoting female reproductive structures and bisexual cone initiation. Similar upregulation of indole-3-acetic acid (IAA)-related genes was observed in bisexual cones of *P. massoniana*[20]. The important role of auxin in initiating bisexual cones may lie in its ability to mediate cell fate reprogramming in diverse pathways, including biosynthesis, polar transport, and signal transduction[69–71]. Studies on *Arabidopsis* have revealed that disturbances in auxin biosynthesis, transport, or signaling lead to pistil development defects[72–75]. Specially, auxin activates *AUXIN RESPONSE FACTOR5/MONOPTEROS* (*ARF5/MP*), *AUXIN AINTEGUMENTA* (*ANT*), and *AINTEGUMENTA-LIKE6/PLETHORA3* (*AIL6/PLT3*), enabling their binding to the *LFY* promoter and directly inducing *LFY* expression[76,77]. *LFY*, in turn, activates *AP3* and *AG* genes in *Arabidopsis*[78,79] and potentially regulates B-class genes in gymnosperms[50]. Thus, the auxin-*LFY* module likely plays a crucial role in initiating the bisexual cones in *P. crassifolia*, as evident by the consistent correlation between *LFY* gene expression gradient and IAA concentration gradient (Figs. 3b, 7 Supplementary Figs. 4a, 10h). Recent studies indicated that sex transition in angiosperms is related to other hormones. In *Cucumis melo*, mutations in two genes involved in ethylene biosynthesis pathway, *CmACS7* and *CmACS11*, lead to stamen degeneration. *Cmacs-7* mutants exhibit an andromonoecious phenotype and *Cmacs11* mutants display an androecious phenotype[80,81]. However, except IAA, there were no significant differences in the content of other hormones among BM, BF, F, and M during cone development (Supplementary Fig. 10h and Supplementary Data 10). In contrast to gymnosperms where the transition from unisexual to bisexual cones, sex transition (from unisexual to bisexual) of *C. melo* results from the failure of programmed abortion of pistil or stamen primordia[80], which may be the reason why their developmental mechanisms are completely different.

Remarkably, the formation of unisexual male cones in Gnetales closely mirrors the development of male flowers in angiosperms. Briefly, in *Ephedra*, *Gnetum*, and *Welwitschia*, female and male

primordia emerge simultaneously in a single cone, with the male reproductive structure being fertile and functional, while the female counterpart undergoes programmed abortion during development[82,83]. Despite extensive research on the mechanism of unisexual flower formation in angiosperms[31,81,84,85], little attention has been directed towards understanding the mechanism underlying unisexual male cone formation in Gnetales. Morphological studies indicate not only the similarity in the formation processes of male reproductive organs between Gnetales and angiosperms but also parallel evolution of some morphological characters, such as broad leaves with netted veins and vessels in *Gnetum* and the absence of archegonia in reduced female gametophytes in *Gnetum* and *Welwitschia*[86,87]. Additionally, phylogenomic studies revealed the occurrence of molecular convergent evolution between Gnetales and angiosperms[4]. Consequently, a comparative investigation into the developmental mechanisms of male reproductive organs in Gnetales and angiosperms promises to provide further insights into the origin of bisexual flowers.

## Conclusion

In the present study, we utilize multi-omics techniques to unravel the mechanisms driving bisexual cone initiation in *Picea crassifolia*. First, a comprehensive analysis of transcriptome and MADS-box gene expression profiles reveals a "bisexual" pattern in bisexual cones. Specifically, the transcriptome patterns of the female and male structures of bisexual cones resemble those of female and male cones, respectively. Combined with the higher expression of C-class genes and *PcLFY* in female structures of bisexual cones and the opposite trend of B-class genes, the developmental mechanism of bisexual cones supports the out of male model. Second, *GGM7* genes are essential for the initiation of the female reproductive structures of bisexual cones. One *GGM7* member, *PcDAL21*, exhibits an expression pattern opposite to B-class genes in unisexual and bisexual cone development. This suggests that the ectopic expression of *PcDAL21* and its reciprocal interaction with B-class genes might influence the development of macrosporophylls or microsporophylls. Another member, *PtDAL10*, extensively interacts with other MADS-box genes in *P. tabuliformis*[36] and *PcDAL10* demonstrates high expression levels in bisexual cones of *P. crassifolia*. Third, in contrast to unisexual cones, bisexual cones exhibit CG and CHG DNA methylation reconfiguration. This reconfiguration potentially influences the expression patterns of some cone development-related genes including *PcDAL12*, *PcDAL10*, *PcNEEDLY*, and *PcHDG5*, which may be one of the key factors in the initiation of bisexual cones. Finally, we observed an auxin concentration gradient in bisexual cones, accompanied by heightened expression of auxin signal pathway genes in the female reproductive structures of bisexual cones. This observation implies a potential significant involvement of auxin in the initiation of bisexual cones.

Based on our multi-omics study, we propose a model explaining the molecular mechanism underlying bisexual cone initiation. We hypothesize that some individuals of Qinghai spruce may undergo DNA methylation reconfiguration and auxin concentration alterations during male cone development. These changes facilitate shifts in *LFY* and *NLY* expression, leading to the establishment of an expression gradient along reproductive axis, with the highest expression at the apex. Upon reaching specific expression thresholds for *LFY* and *NLY*, the initiation of *DAL21* expression occurs. Due to the functional antagonism between *DAL21* and B-class genes, coupled with the regulatory influence of DNA methylation, the expression of B-class genes becomes progressively weaker and ultimately inactive. Consequently, this results in the changes of sex-determining protein complexes at the tip of the reproductive axis, thereby promoting megasporophyll development. In this intricate process, *DAL10*, whose expression is upregulated in bisexual cones through methylation or regulation of *LFY* and *NLY* genes, plays a pivotal role in shaping the protein interaction network during sex determination (Fig. 7).

Our study identifies some key factors, including DNA methylation reconfiguration, reciprocal expression of MADS-box genes, and a gradient in IAA concentration, that significantly contribute to the formation of hermaphrodite in gymnosperms. Nevertheless, several challenges remain in the investigation of bisexual cones: (1) the lack of genomic data for *P. crassifolia* and inadequate depth of methylation sequencing hinder a comprehensive profiling of methylation patterns; (2) it is difficult to distinguish bisexual cones from unisexual cones and obtain experimental samples before the initiation of bisexual cones; and (3) the role of auxin in bisexual cone development requires further exploration due to the limited existing research on this topic. Considering these limitations in materials and methodologies, further investigation is still needed to deepen our understanding of the molecular mechanisms governing bisexual cone initiation in gymnosperms.

## Materials and methods

**Sample collection**. Two trees of *Picea crassifolia* Kom. were selected from the China National Botanical Garden in Beijing, China (116°12′33.49″E, 39°59′26.87″N), including a "normal tree" producing unisexual cones only yearly in March–April and a "bisexual cone tree" exhibiting a consistent bisexual cone phenotype yearly (Fig. 1a–c). In March-April, male and female cones at five developmental stages (M1–5 and F1–5) of the "normal tree" and bisexual cones at three developmental stages (BM4-6 and BF4-6) of the "bisexual cone tree" were collected (Fig. 2a). The male and female structures of bisexual cones were separated using blades. Transition zones were eliminated for data reliability. All samples were immediately frozen in liquid nitrogen and stored at −80 °C or fixed in FAA.

**Pollen germination and statistics**. We compared pollen germination rates between normal and bisexual cones, excluding a parallel assessment of seed activity due to the limited availability of seed-producing bisexual cones. This constraint stems from the gradually drop of most bisexual cones after the pollen dispersion from their male structures. Although some bisexual cones exhibited apparent seed production, all seeds aborted. Pollen grains were collected from six normal male cones (stage M5) and six bisexual cones (stage BM5) and cultured in medium containing 18% sucrose and 0.01% boric acid at 25 °C for 48 h. A count of approximately 300 pollen grains were performed for each group, and a chi-square test was employed to assess the statistical difference between the two groups.

**Transcriptome sequencing and analysis**. RNA sequencing was performed on male, female, and bisexual cones of each stage (M1-M5, F1-F5, BM4-5, and BF4-5). In addition to two biological replicates at each developmental stage of the female cones, three biological replicates were used for each developmental stage of the other three tissues, and a total of 37 transcriptomes were obtained. Detailed information about these transcriptomes is shown in Supplementary Table 4. The experimental procedures were conducted as follows: (1) Total RNA was extracted from each sample using the RNAprep Pure Plant Kit (TIANGEN, Beijing, China); (2) A minimum of 5 µg of RNA per sample was used to construct a cDNA library following the manufacturer's

instructions for NEBNext® Ultra™ Directional RNA Library Prep Kit for Illumina® (New England BioLabs Inc., Ipswich, MA, USA); (3) The cDNA library was quantified using an Agilent Bioanalyzer 2100 (Agilent Technologies, Santa Clara, CA, USA) and real-time fluorescence quantitative PCR system; (4) Sequencing of all cDNA libraries were carried out on the Illumina Novaseq platform using a 150 bp paired-end sequencing strategy, resulting in 5–6 Gb of raw reads per library.

To remove adaptors and low-quality reads, raw reads were filtered using Trimmomatic v0.38[88]. Clean reads were then mapped against the *P. abies* genome[89] using HISAT2 v2.0.5[90], and genome-aligned transcripts were assembled for each sample with StringTie v1.3.3[91] to calculate the expression level of each gene. Transcript levels were normalized using the transcripts per million (TPM) parameter. Next, differential expression analysis employed DEseq2 v1.34.0[92], with significance assigned to genes exhibiting FDR ≤ 0.05 and |log2fold change| ≥1. GO (Gene Ontology) and KEGG (Kyoto Encyclopedia of Genes and Genomes) enrichment analyses were conducted using Omics-Share (https://www.genedenovo.com/). Pearson correlation coefficients and principal component analysis were executed using the R packages Corrplot and Ade4, respectively.

**Bisulfite sequencing (BS-seq) and analysis**. Given the pivotal role of the fourth developmental stage in bisexual cone initiation, BS-seq libraries were constructed for various tissues (M4, F4, BM4, and BF4) from this stage, each represented by three biological replicates. Total genomic DNA was extracted from each sample using a modified CTAB method[93]. Prior to bisulfite treatment, unmethylated Lambda DNA was added to the sample as a control, and Covaris M220 was used for fragmentation, with a target peak at 300 bp. Construction of the BS-seq library employed the NEBNext Ultra DNA Library Prep Kit for Illumina (NEB #E7370), using at least 100 ng fragmented DNA per sample. Adapter ligation involved NEXTflex™ Bisulfite-Seq Barcodes – 6 (Illumina Compatible, Catalog #511911) as methylated adapters. Bisulfite treatment utilized the MethylCode™ Bisulfite Conversion Kit (Catalog no. MECOV-50), and PCR amplification was performed using EpiMark® Hot Start Taq DNA Polymerase (NEB #M0490). Subsequently, the BS-seq libraries were sequenced on the Illumina HiSeq X Ten platform, yielding a minimum of 33 GB of clean reads for each sample. Detailed information on the methylomes is shown in Supplementary Table 2.

The methylome analysis comprised several steps. First, BS-seq clean reads were aligned to λ-DNA using Bismark v0.23.0 software to determine the bisulfite nonconversion rate[94]. Second, a RefSeq database was constructed by merging and de novo assembly of clean reads from six transcriptomes (M1, M3, M5, F1, F3, and F5) using Trinity v20140717[95]. Redundant transcripts were removed with CD-HIT v4.6.5[96], coding sequences were identified using TransDecoder v20140704[97]; and the longest transcript for each gene was selected with a custom Perl script[98]. Third, BS-seq clean reads were mapped to the RefSeq database, and duplicate reads were removed using Bismark. Methylation calling for each cytosine was conducted, calculating the methylation level by dividing the number of reads supporting methylated cytosine by the total number of reads covering that cytosine. A binomial test with a $P$ value < 0.01, based on the estimated nonconversion rate, identified methylated cytosine sites. Finally, body-methylated genes were defined following the method of Takuno and Gaut[99].

To identify differences in DNA methylation among tissues, we identified DMCs and DMRs by comparing methylation levels at individual cytosine site. CGmapTools v0.1.1 was used to define DMCs as cytosine exhibiting a methylation level difference ≥0.1

with a $P$ value < 0.05 in the chi-square test[100]. Detection of DMRs between tissues employed Metilene v0.2-8, with a minimum mean methylation difference set to 0.1[101]. In addition, we calculated differential methylation levels of genes between tissues, identifying significantly differentially methylated genes as those with methylation level differences ≥0.1 and $P$ values < 0.05 in the chi-square test.

We analyzed promoter DNA methylation, a crucial gene regulation mechanism in plants[102], focusing on MADS-box genes, *PcLFY*, *PcNLY*, and *PcHDG5*. Using the genome sequences of *P. abies*[89] and *Picea glauca*[103] as references, we obtained the ~2000 bp upstream region of these genes in *P. crassifolia* and then extracted the DNA methylation information for these upstream regions using the same method. The primers used for amplifying the promoter sequences are shown in Supplementary Data 11.

**Gene family identification and phylogenetic reconstruction**. To identify MADS-box sequences, we employed two Pfam models, SRF (PF00319) and K-box (PF01486), with HMMER v3.3.2 software ($E$ value < 1e-5), searching protein sequences in the RefSeq database[104]. All sequences were manually checked using the NCBI conserved domain database (http://www.ncbi.nlm.nih.gov/Structure/cdd/wrpsb.cgi) and InterProScan program (https://www.ebi.ac.uk/interpro/) to ensure the accuracy of candidate proteins. The same approach was applied to identify MADS-box sequences in the genome data for *Amborella trichopoda*, downloaded from Ensembl_ftp: http://ftp.ensemblgenomes.org/pub/plants/release-54/fasta/amborella_trichopoda/. For phylogenetic analysis, MADS-box genes from *Arabidopsis thaliana* and *Oryza sativa* subsp. *Japonica* were retrieved from PlantTFDB (http://planttfdb.cbi.pku.edu.cn/index.php). Protein sequences of MADS-box genes from *P. abies* and *Pinus taeda* were obtained from NCBI and the *P. taeda* genome, respectively, as reported in previous studies[44,105]. Additionally, MADS-box gene sequences from *Gnetum luofuense* and *Ephedra equisetina*, as reported by Hou et al.[106], were included in our phylogenetic analysis.

The gene *PcHDG5*, a member of the HD-ZIP IV subfamily, exhibited significant differences in the integrated analysis of transcriptome and methylome data (see Results and Discussion for details). Consequently, a phylogenetic analysis of the HD-ZIP IV subfamily was conducted. HMMER software ($E$ value < 1e-5) was employed with Homeodomain (PF00046) and START (PF01852) Pfam models to search HD-ZIP IV subfamily genes in the RefSeq database, and protein sequences from *P. abies*[89] and *Welwitschia mirabilis*[34] were utilized. Confirmation of HD-ZIP IV domains used the Conserved Domain Database of NCBI and the Inter Pro Scan program. Protein sequences of HD-ZIP IV subfamily genes from *Oryza sativa* and *Arabidopsis thaliana* were retrieved from PlantTFDB according to previous studies[107,108]. Detailed information is provided in Supplementary Data 1 and 9. Alignment of all MADS-box gene sequences and HD-ZIP IV gene sequences utilized MAFFT v7.453[109], and alignment uncertainty was calculated using ZORRO[110] with a threshold of 5. Subsequently, RAxML v8.2.11[111] was used to construct a maximum-likelihood (ML) tree for each gene family with the PROTGAMMAAUTO model and 1000 bootstrap replicates.

**Quantification of gene expression levels and qRT−PCR analysis**. Clean reads for each sample were mapped to the RefSeq database using Bowtie2 v2.2.5[112], and gene expression levels (TPM) were calculated using RSEM v1.2.15[113]. DEseq2 facilitated differential expression analysis, and the expression levels of all MADS-box and HD-ZIP IV genes, as well as *PcLFY* and *PcNLY*, were extracted. Subsequently, we sampled the male (M1-5), female (F1-5) and bisexual (BM4-6 and BF4-6) cones (Fig. 2a) used for qRT−PCR to

verify the expression differences of some genes, including 6 MADS-box genes, 1 HD-ZIP IV gene, *PcLFY*, *PcNLY*, and 14 genes associated with de novo methylation, maintenance methylation, and demethylation processes. The primer details for qRT-PCR are provided in Supplementary Data 11.

**RNA in situ hybridization**. RNA in situ hybridization was used to detect the expression patterns of *PcAP3/PI* and *PcDAL21*, mitigating potential inaccuracies in RNA-seq and qRT-PCR results arising from imprecise sampling. B4 (the whole bisexual cone) from the "bisexual cone tree" and F4 from the "normal tree" were collected for RNA in situ hybridization. Gene-specific sense and antisense probes (200–300 bp) were synthesized using $T_7$ RNA polymerase and labeled with digoxigenin via a DIG RNA Labelling Kit (SP6/T7; Roche, Mannheim, Germany). Hybridization was carried out at 50 °C for 16 h with 8–10 ng/µl RNA probes. Following the protocol by Brewer et al.[114] with minor modifications, all steps for in situ hybridization were performed. Microscope images were captured using a Leica DM6 B microscope (Leica, Wetzlar, Germany). The primers used for in situ hybridization are shown in Supplementary Data 11.

**Hormone measurement**. Hormone contents in three biological replicates of male cones (M3-5), female cones (F3-5), and bisexual cones (BM4-5 and BF4-5) were quantified, with a minimum of 5 buds per biological replicate. Purification of ~100 mg of frozen plant material samples utilized C18 reversed-phase, polymer-based, solid-phase extraction (RP-SPE) cartridges, followed by analysis on a UPLC– Orbitrap-MS system (UPLC, Vanquish; MS, QE) and a Q Exactive hybrid Q–Orbitrap mass spectrometer equipped with a heated ESI source (Thermo Fisher Scientific Inc., USA). Data acquisition employed Q-Exactive with Xcalibur 4.1 (Thermo Fisher Scientific Inc., USA) and processing utilized TraceFinder™4.1 Clinical (Thermo Fisher Scientific Inc., USA). All experiments were conducted at BioNovoGene (Suzhou, China). The measured hormone concentrations included IAA, salicylic acid (SA), gibberellins (GA1, GA3, GA4, GA7), abscisic acid (ABA), jasmonic acid (JA) and jasmonic acid-isoleucine (JA-Ile).

**Dual-luciferase assay**. Given the differential methylation and expression observed in both M4 vs. F4 and BM4 vs. BF4, dual-luciferase reporter assays (dual-LUC) were employed to verify interaction between *PcHDG5* and *PcAP3/PI*. The specific steps were as follows. The effector was created by amplifying and cloning the CDS of *PcHDG5* into the Super1300-GFP vector. Upstream sequences of *PcAP3/PI* (2000–3000 bp) were amplified and individually cloned into the pGreen II 0800-LUC vector as reporters. However, due to limited reference sequences (*P. abies* and *P. glauca* genomes), only a 650-bp upstream sequence of *PcDAL11* (c124395_g1_i1) could be obtained and was excluded from the dual-luciferase assay. The effector and reporters were separately transformed into *Agrobacterium tumefaciens* (GV3101), with the culture adjusted to $OD_{600} = 0.6$–0.8. Each pair of effector and reporter was mixed at a 1:2 ratio and then injected into at least three leaves of *Nicotiana benthamiana*, with the reporter as the control. Following a 48-h growth period at 23 °C, LUC and REN activities were measured using the Dual-luciferase Reporter Assay System (Promega, Madison, USA) according to the manufacturer's instructions.

**Statistics and reproducibility**. RNA-seq employed three biological replicates for male cones (M1–M5), male structures of bisexual cones (BM4–BM5), and female structures of bisexual cones (BF4–BF5), with two biological replicates for female cones (F1-F5).

QRT-PCR analysis involved three biological replicates for all tissues (M1–M5, F1–F5, BM4–BM6 and BF4–BF6). Methylome analysis utilized three biological replicates for each tissue (M4, F4, BM4, and BF4), where each cone represented one biological replicate. Hormone measurements were conducted with three biological replicates, each comprising a minimum of five buds.

In the Dual-luciferase assay, LUC and REN activities were measured using four injected tobacco leaves. Statistical tests included Student *t*' test and chi-square test using R v4.1.2, the two-tailed Wilcoxon test by Python v2.7.14, and One-way ANOVA analysis using OmicsShare tools.

**Reporting summary**. Further information on research design is available in the Nature Portfolio Reporting Summary linked to this article.

## Data availability

All raw data generated in this study are deposited in the SRA database. The BioProject accessions for RNA-seq data and methylome data are PRJNA981226 and PRJNA983433, respectively, with corresponding SRA accession numbers for each sample detailed in Supplementary Tables 2 and 4. The numerical source data supporting both main and supplementary figures can be found in Supplementary Data 12. The reference sequence, upstream regions obtained by PCR, protein sequence matrices of MIKCc MADS-box genes, and the ML tree of the MIKCc MADS-box gene family are deposited in Drayd Digital Repository[115] (https://doi.org/10.5061/dryad.dbrv15f6w).

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

## Acknowledgements

We sincerely thank Dr. Shihui Niu from Beijing Forestry University for providing expression data of *Pinus tabuliformis* cones, and Dr. Pichang Gong and Yang Dong from Institute of Botany, Chinese Academy of Sciences for their kind help in in situ hybridization experiment and dual-luciferase assay. This study was supported by Key Research Program of Frontier Sciences, CAS (QYZDJ-SSW-SMC027), the Strategic Priority Research Program, CAS (grant no. XDA23080000), the National Natural Science Foundation of China (grant no. 31770238), and the K.W. Wong Education Foundation (grant no. GJTD-2020-05).

## Author contributions

X.-Q.W. and J.-H.R. conceived and supervised the project. Y.-Y.F. and H.D. collected samples and performed experiments. Y.-Y.F., H.D. and K.-Y.H. performed data analysis. Y.-Y.F., J.-H.R. and X.-Q.W. wrote the manuscript. All authors read and approved the article.

## Competing interests

All authors declare no competing interests.

## Additional information

**Peer review information** : *Communications Biology* thanks Israel Ausin, Qing-Feng Wang and the other, anonymous, reviewer(s) for their contribution to the peer review of this work. Primary Handling Editor: David Favero. A peer review file is available.

