## [Peer Review File · Communications Biology]

Reviewers' comments:

Reviewer #1 (Remarks to the Author):

The manuscript presented by Feng et al., 2023 shows the results of studying one of the most outstanding questions in botany and evolutionary biology: how the bisexual flowers in modern angiosperms evolved from previous gymnosperms' ancestors. This question has been addressed for over a century now. Feng et al., 2023 investigate this question by applying a combination of transcriptomics and analyzing the methylome of spontaneously occurring aberrant bisexual cones in *Picea crassifolia*, a specie that typically bears only female or male cones only.

From the analysis of the expression of B-class in male, female, and bisexual cones, it follows that out of the three main theories (mostly male, out of female, and out of male) that aim to explain the origin of bisexual flowers, the "out of male" theory is the most consistent with their findings. Moreover, the authors also offer a possible explanation for the differences observed in some differentially expressed genes. Differences in DNA methylation might cause differential expression. Thus, this article represents a step ahead in explaining the molecular mechanisms that led to bisexual flowers.

The paper's scientific quality is generally good and does read smoothly, although the manuscript would benefit from light editing. The figures and presentations are more than correct. However, the reading of the paper raises a few questions of minor importance:

1. in the line 30

"we reveal the molecular mechanism of bisexual cones in the conifer *Picea crassifolia*"

and line 37

"This study unveils the mechanisms responsible for bisexual cone formation in conifers and provides new clues to the development of bisexuality in the origin of flowers"

The tone could be softened, considering that the authors are showing a correlation, not strictly causation, and only two individuals of one species had been considered. This being said, I agree that the data are compelling; These could be one of the mechanisms, and there might be others.

2. line 200. I believe MRCA hasn't been spelled out before in the text

3. line 330-336. The RdDM pathway doesn't have any effect on the maintenance of CG or CHG methylation once they are established; these two marks are deposited by MET1 and CMT3, respectively. The "weakening" of the RdDM pathway couldn't explain a reduction in the level of CG and CHG methylation; only lack of maintenance or active demethylation could.

Reviewer #2 (Remarks to the Author):

Review of Feng et al. manuscript termed 'Expression trade-off of MADS-box genes and DNA methylation reconfiguration initiate bisexual cones in spruce'.

The inclusion of DNA methylation patterns in the study of bisexual cone development is useful, but it does not provide the missing link to understanding the development of true (hermaphrodite) flowers in angiosperms, as the authors imply in a few places in their manuscript. In addition, the current manuscript version is lacking the depth to provide new insights. It is not clear what the authors want

to imply by trade-off in their manuscript title.

The Discussion is rather short (not even 2 pages), while the Results section is too long (12 pages). The authors are already discussing results in the Results section, but this would not be appropriate. Also, the authors should not be referring so much to Figures and Supplementary Figures in their short discussion.

Thus, this manuscript needs to be thoroughly reworked to communicate results and their broader context better.

Further comments:

Line 35: 'some important genes': which, how many?

Line 47: I don't think it is true. There exist dioecious angiosperms.

Lines 67-70: This statement needs the appropriate references.

Line 69: Has it indeed been useful to explain the origin of flowers, how precisely?

Line 70: If these cones are aberrant, why they can occur widely? One would assume there is natural selection against aberrant structures.

Line 74-77: I think that the summary of their results could be more explicit.

Line 79-82: My understanding is that male and female structures in bisexual cones were separated immediately after collection (Niu et al. 2016). What the actual shortcoming here was, was that not early enough developmental stages of bisexual cones were examined in their study.

Line 88: What do you mean by various methylated regions? Can you be more explicit?

Line 93: What are those genes named for?

Line 94-96: I think the work by Yakovlev et al. 2020 could be cited here, cf.

https://link.springer.com/chapter/10.1007/978-3-030-21001-4_5.

Line 100-102; 116-117: Perhaps conclusions cannot be as far reaching at this point as to the origin and evolution of (true) flowers in angiosperms, see also your Lines 62-66 which actually provide evidence.

Line 104: Is it a monoecious species? Please add this information here.

Line 105-107: how are these results different from the ones obtained by Niu et al. 2016?

Line 105: I think you should introduce to the reader what B-class and C-class genes are and provide examples for those whose precise function is known. There is not much (concise) information about the evo-devo on reproductive development in seed plants.

Line 108: please remove: 'Integrated with DNA methylomic analysis' (it is redundant information to what is described next).

Line 110-112: It is interesting that you mention PcNEEDLY (only expressed in female cones usually) as an example, since this gene has no counterpart in angiosperms.

Line 113-114: Auxin has already been implicated in archegonial development (related to the female gametophyte). Some background information on such research could be provided here.

Line 116-117: how does this study provide new clues on hermaphroditic flowers?

Lines 67, 70, 384: The term "abnormal" or aberrant in the reproductive development of gymnosperms has never been explained here, in the sense of how often these structures occur and what is the reason for their occurrence. This is perhaps the most intriguing question.

Line 124-126: to my knowledge NLY has no counterpart in angiosperms thus cannot help to explain there the appearance of hermaphroditic (bisexual) flowers. Or is this not what was meant here to describe?

Line 377-379: this is definitely not part of a Results section!

Line 382-390: This is redundant to previous information provided.

Line 384-385: In which sense are those bisexual cones resembling primitive flowers? Something about homology and analogy of reproductive structures should have been introduced and discussed.

Line 390-397: How are those results different from Niu et al., 2016?

Line 398: Please explain somewhere in the manuscript the exact functional mode of action of GGM7 genes and provide the explanation of the abbreviation for the GGM7 genes. Also, it is not clear from your writing what is the new knowledge from your study and what is already known (a reference in your discussion).

Line 405-408: A lot more information on GGM7 genes' function is needed in your discussion to more fully grasp their exact importance in flower evolution.

Line 406: No previous introduction about E-class genes was provided.

Line 410: Can you provide more information about the exact methylation reconfiguration that was observed and its implication? And without having to consult figures and supplement material?

Line 411-414: ditto

Line 415: In my opinion, the function of localized auxin implication should be more emphasized for this research.

Line 417-420: The exploration of the mechanism of bisexual cone initiation is unclear unless a mechanistic model can be developed on the study's results. And this should be better highlighted here.

Figures 1a-c and 2a, 3b, 3e, 3f and 7: It would be important to indicate the developmental stage of those cones, and in all cases. All those codes are not explained in the figure legends.

Figure 3e,f: While an in situ hybridisation experiment is worthwhile, it is very difficult to properly discern the localization in those images.

Figure 4a,b, f(right plot): Have any statistical tests been performed on those patterns?

Figure 5a: Are those patterns significantly different?

Figures 6, 7: contain a lot of information. But I think the overall outline could be improved. For example, the reporter/effector study of the DAL13 promoter could be merged with the in situ localization results (which should become more evident), and the legend for Figure 7 needs to be improved. I didn't find the description very clear.

Reviewer #3 (Remarks to the Author):

This manuscript performed transcriptomic, DNA methylomic, and metabolomic analyses to explore the molecular mechanism of the bisexual cone initiation in *Picea crassifolia*. The authors concluded that the developmental mechanism of bisexual cones is consistent with the out of male model and MADS-box family genes and their regulated genes are essential for the initiation of the female reproductive structure of bisexual cones. They also found that the expression patterns of some cone development related genes might be affected by the DNA methylation variation and the production of the female reproductive structure of bisexual cones might be closely related to the auxin content variation in the initiation of bisexual cones. The study makes a contribution to the molecular mechanism of the bisexual cone initiation in conifers and the evolution of flowers of angiosperms.

Some general comments conducted:

1. Expression profiles of the male structure of bisexual cones are more similar to that of male cones in both *P. crassifolia* and *Pinus tabulaeformis*, which is not consistent with the results mentioned in Niu et al (2016) (Figure 2b,c; Supplementary Figure 2; P7 L146-147; P7 L159- P8 L161). The authors claimed that the reason for this inconsistency might be due to the deviation of analysis methods (P8 L161-162). In addition to Pearson correlation coefficients and principal component analysis (PCA)

performed in this study, more convincing analysis methods are needed to confirm the results.

2. The authors found that F4 vs. BF4 has the least number of differentially expressed genes among M4, F4, BM4, and BF4 (Figure 2d; P7 L148-150). It appears that female cones of bisexual cones are more likely to arise from female cones of unisexual cones than from male cones of unisexual cones. However, the authors stated that the out of male model hypothesis could be used to explain the origin of bisexual cones of *P. crassifolia* after comparing the expression patterns of MIKCC-type MADS-box genes, LFAFY, and NEEDLY and performing six years of field observation (Figure 3; Supplementary Figures 3, 4). A comment on this conflict is needed.

3. PchDG5 can bind to the promoters of PcDAL12, PcDAL13, and c113171_g1_i1 (Figure 6b; Supplementary Figure 7b,c; P13 L284-288). The authors concluded that PchDG5 might activate transcription of downstream MADS-box genes by binding to the core-binding motif (P13 L288-290). Do the authors know which motifs PchDG5 directly binds to?

4. Both the auxin content and the expression of auxin signal related genes differ among the different stages of unisexual cones and bisexual cones (Supplementary Figure 10). How does the auxin signal pathway couple with MADS-box gene related regulation pathway at the initiation stage of bisexual cones? Also, it is better to add the GO enrichment analysis of DEGs in M4 vs. BF4.

5. Figures 1c, 2a, 4b and Supplementary Figure 1c,f are not referred in the main text.

6. The abbreviations should be stated in the figure legend of Supplementary Figure 2.

7. It seems that Figure 4a, b, and c are the same with Supplementary Figure 6.

Response to reviewers

Reviewers' comments:

Reviewer #1 (Remarks to the Author):

The manuscript presented by Feng et al., 2023 shows the results of studying one of the most outstanding questions in botany and evolutionary biology: how the bisexual flowers in modern angiosperms evolved from previous gymnosperms' ancestors. This question has been addressed for over a century now. Feng et al., 2023 investigate this question by applying a combination of transcriptomics and analyzing the methylome of spontaneously occurring aberrant bisexual cones in *Picea crassifolia*, a specie that typically bears only female or male cones only.

From the analysis of the expression of B-class in male, female, and bisexual cones, it follows that out of the three main theories (mostly male, out of female, and out of male) that aim to explain the origin of bisexual flowers, the “out of male” theory is the most consistent with their findings. Moreover, the authors also offer a possible explanation for the differences observed in some differentially expressed genes. Differences in DNA methylation might cause differential expression. Thus, this article represents a step ahead in explaining the molecular mechanisms that led to bisexual flowers.

The paper's scientific quality is generally good and does read smoothly, although the manuscript would benefit from light editing. The figures and presentations are more than correct.

>>>Thanks for the positive comments.

Minor questions:

1. in the line 30

“we reveal the molecular mechanism of bisexual cones in the conifer *Picea crassifolia*”

and line 37

“This study unveils the mechanisms responsible for bisexual cone formation in conifers and provides new clues to the development of bisexuality in the origin of flowers”

The tone could be softened, considering that the authors are showing a correlation, not strictly causation, and only two individuals of one species had been considered. This

being said, I agree that the data are compelling; These could be one of the mechanisms, and there might be others.

>>>Thanks for the insightful comment and reminder. In the revision, these two sentences have been modified to:

“Here, we employed transcriptomic and DNA methylomic analyses, together with hormone measurement, to investigate the molecular mechanisms underlying bisexual cone development in the conifer species *Picea crassifolia*”(Lines 35-38).

“This study unveils the potential mechanisms responsible for bisexual cone formation in conifers and may shed light on the development of bisexuality” (Lines 44-45).

2. line 200. I believe MRCA hasn't been spelled out before in the text

>>>It has been spelled out in the revision. MRCA is the abbreviation of most recent common ancestor. **Line 458.**

3. line 330-336. The RdDM pathway doesn't have any effect on the maintenance of CG or CHG methylation once they are established; these two marks are deposited by MET1 and CMT3, respectively. The “weakening” of the RdDM pathway couldn't explain a reduction in the level of CG and CHG methylation; only lack of maintenance or active demethylation could.

>>> *De novo* methylation, maintenance methylation, and demethylation are all crucial for DNA methylation variation. DNA methylation in all sequence contexts is established by the RdDM pathway. Consequently, changes in the expression of genes associated with the RdDM pathway could impact DNA methylation levels in all sequence contexts, which has been confirmed by many previous studies. For instance, in rice, the homozygous *OsDRM2* (-/-) disruptant exhibits large reductions in methylation levels at symmetric sites (CG and CHG) and a notably lower percentage of cytosine methylation in asymmetric CHH sequences when compared to wild-type plants (Moritoh et al., Plant J. 2012, 71:85-98). Similarly, in *Arabidopsis*, *UVR8* interacts with and negatively regulates *DRM2* by preventing its chromatin association and inhibiting the methyltransferase activity, then affecting DNA methylation in all sequence contexts (Jiang et al., Nat. Plants, 2021, 7:184-197). In addition, Cheng et al. (2018) found that several genes encoding DNA methyltransferases and other key components in the RdDM pathway were significantly downregulated during strawberry fruit ripening. They inferred that the reduction of RdDM activity during strawberry ripening, such that DNA demethylation becomes relatively dominant over methylation, lead to an overall loss of DNA methylation (Cheng et al., Genome Biol., 2018, 19:212).

In the present study, the expression levels of most genes related to maintenance methylation and demethylation pathways, as well as four *de novo* methylation-related genes, are generally higher in F4 compared to M4. However, three *de novo* methylation-related genes in M4 exhibited significant upregulation compared to F4,

potentially explaining the similar DNA methylation levels in both tissues. In BF4, genes related to *de novo* methylation, maintenance methylation, and demethylation pathways exhibited a similarity to F4. However, two genes involved in *de novo* methylation, *AGO4* and *MORC6*, showed significant expression differences between F4 and BF4, and most genes related to the RdDM pathway were expressed at higher levels in BF4 than in the other three tissues, suggesting a potentially heightened activity of the RdDM pathway in BF4, which could contribute to increased CG and CHG methylation in BF4. In contrast, BM4 showed significant downregulation of three genes associated with the RdDM pathway compared to M4, although *MET1*, *CMT3*, and *DDMI* in BM4 were upregulated compared to M4. Such weakening of the RdDM pathway in BM4 might result in a reduction in CG and CHG methylation levels, although the high expression of *MET1* and *CMT3* could enhance the maintenance of CG and CHG methylation (Fig. 6e and Supplementary Fig. 9). However, further investigations are necessary to understand the reasons behind the lack of significant effects of RdDM changes on CHH methylation in *Picea crassifolia*.

In the revision, we have added information as follows:

“The dynamic regulation of DNA methylation might be associated with changes in expression levels of genes related to *de novo* methylation, maintenance methylation, and demethylation. Particularly, the RdDM pathway, initiating *de novo* methylation, affects methylation broadly across all sequence contexts (Moritoh et al., Plant J. 2012, 71:85-98; Jiang et al., Nat. Plants, 2021, 7:184-197; Cheng et al., Genome Biol., 2018, 19:212).” **Lines 860-864.**

“Comparing methylation-related gene expression across tissues revealed similarities in *de novo* methylation, maintenance methylation and demethylation genes between BF4 and F4. The majority of DNA methylation-related genes show higher expression in BF4 than BM4, implying methylation process activation in BF4. Most genes involved in the RdDM pathway expressed at higher levels in BF4 than in F4, with significant differences in *AGO4* and *MORC6*, suggesting active RdDM pathway in BF4, which resulted in elevated CG and CHG methylation levels (Fig. 6e and Supplementary Fig. 9). In contrast, the RdDM pathway weakened in BM4 compared to M4, possibly reducing CG and CHG methylation, although upregulation of *MET1* and *CMT3* might enhance the ability to maintain these methylations.” (**Lines 864-909**).

Reviewer #2 (Remarks to the Author):

Review of Feng et al. manuscript termed 'Expression trade-off of MADS-box genes and DNA methylation reconfiguration initiate bisexual cones in spruce'.

The inclusion of DNA methylation patterns in the study of bisexual cone development is useful, but it does not provide the missing link to understanding the development of true (hermaphrodite) flowers in angiosperms, as the authors imply in a few places in their manuscript. In addition, the current manuscript version is lacking the depth to

provide new insights. It is not clear what the authors want to imply by trade-off in their manuscript title.

>>>Thanks for the reminder. The revised manuscript has focused on the mechanisms underlying the development of bisexuality in conifers rather than the development of true flowers in angiosperms. Moreover, we have combined the Results and Discussion into a single section and rewritten this section very carefully.

(1) Flores-Rentería et al. (Am. J. Bot., 2013, 100:602-612) found that sexual inconstancy was only detected in some unisexual individuals, and suggested that genetic plasticity might contribute to the formation of complex sexual systems in gymnosperms. Interestingly, in the China National Botanical Garden in Beijing, we observed morphological difference of cones between two adjacent trees of *Picea crassifolia*. One tree, referred to as the “normal tree”, consistently produces unisexual cones yearly in March-April. In contrast, the other tree, named as the “bisexual cone tree”, consistently exhibits a bisexual cone phenotype every year. Therefore, we hypothesized that genetic plasticity plays a crucial role in the initiation of bisexual cones. This hypothesis is supported by the integrated analysis of DNA methylome and transcriptome data. Our results showed that in contrast to unisexual cones, bisexual cones exhibited CG and CHG DNA methylation reconfiguration, with significant changes in global methylation levels, methylation site distributions, as well as the methylation level differences of DMCs (Fig. 4a-d, 5a-b). Remarkably, the female and male structures of bisexual cones show distinct methylation change patterns (Fig. 4a-d). This variation in DNA methylation has the potential to influence the expression patterns of some important genes involved in cone development, including *PcDAL12*, *PcDAL10*, *PcNLY* and *PcHDG5*, which may be one of the key factors in the initiation of bisexual cones. The expression pattern of *DAL21* (belong to *GGM7* clade, specific expression in female cones) is opposite to that of B-class (specific expression in male cones) genes during the development of unisexual and bisexual cones (Fig. 3b, c). Expression of *DAL21* and B-class genes in the same reproductive axis may be the key to the development of bisexuality. During this process, the expression of *PcDAL12* (B-class) and possible upstream genes of B-class genes, *HDG5*, *DAL10* and *NLY*, may be regulated by DNA methylation, as mentioned above. Based on these information, we have proposed a new model for the initiation of bisexual cone.

(2) In the conclusion, we summarized our findings as follows: “Based on our multi-omics study, we propose a model explaining the molecular mechanism underlying bisexual cone initiation. We hypothesize that some individuals of Qinghai spruce may undergo DNA methylation reconfiguration and alterations in auxin concentration during male cone development. These changes facilitate shifts in *LFY* and *NLY* expression, leading to the establishment of an expression gradient along reproductive axis, with the highest expression at the apex. Upon reaching specific expression thresholds for *LFY* and *NLY*, the initiation of *DAL21* expression occurs. Due to the functional antagonism between *DAL21* and B-class genes, coupled with the regulatory influence of DNA methylation, the expression of B-class genes becomes progressively weaker and ultimately inactive. Consequently, this results in the changes of sex-determining protein complexes at the tip of the reproductive axis, thereby promoting megasporophyll development. In this intricate process, *DAL10*, whose expression is upregulated in bisexual cones through methylation or regulation

of *LFY* and *NLY* genes, plays a pivotal role in shaping the protein interaction network during sex determination (Fig. 7).” (Lines 1103-1140).

(3) The MADS-box genes determine bisexual cone formation by modulating gene expression levels and altering interaction patterns, exemplifying a complex and subtle process characterized by a delicate trade-off.

Based on the ABC model established in *Arabidopsis*, B-class floral homeotic genes express in petals and stamens, with the B + C specifying stamen development. Meanwhile, C-class genes express in stamens and carpels, thus determining the carpel development (Ma & dePamphilis, Cell, 2000, 101:5-8). This model can be extended to gymnosperms, where B + C genes govern male cone development, while B-class genes exclusively express in male reproductive structures, and C-class genes control the formation of female cones (Wang et al., Plant. J., 2010, 64:177-190). According to Theißen et al. (2002), who proposed the out of male and out of female models based on B-class gene expression changes to explain the formation of hermaphrodite, the former suggested that reduced B-class gene expression in the upper part of the male cone leads to ectopic ovule development, while the latter assumed that ectopic expression of B-class genes in the basal part of the female cone results in the ectopic development of male reproductive units.

In the present study, except for B-class genes, the expression patterns of almost all MADS-box genes in the female structures of bisexual cones are similar to those in female cones. Even among B-class genes, except that the expression level of *PcDAL12* in BF4 is slightly higher than that in M4, other members show significantly higher expression levels in BF4 than in F4 but lower than in M4 and BM4 (Fig. 3b, c). These results are supported by *in situ* hybridization, that is, the specific signals of B-class genes occur at the base of female structure and tapetum of male structure in bisexual cones (Fig. 3d and Supplementary Fig. 3).

DAL21, a member of the *GGM7* subfamily (absent in angiosperms), is associated with initiation of the female reproductive development programme (Carlsbecker et al., New Phytol., 2013, 200: 261-275), and displays specific expression in female cones. It shows an opposite expression pattern with B-class genes during the development of unisexual and bisexual cones (Fig. 3b, c). We infer that *PcDAL21* may function in female cones similar to how B-class genes act on male cones, establishing female identity and potentially antagonizing B-class genes. Our findings reveal an opposing expression gradient of *DAL21* and B-class genes during bisexual cone initiation. This expression pattern suggests a trade-off on the same reproductive axis, potentially influencing the development of macrosporophylls or microsporophylls in bisexual cones. This information has been added in the revision. Please see lines 1087-1091.

The Discussion is rather short (not even 2 pages), while the Results section is too long (12 pages). The authors are already discussing results in the Results section, but this would not be appropriate. Also, the authors should not be referring so much to Figures and Supplementary Figures in their short discussion. Thus, this manuscript needs to be thoroughly reworked to communicate results and their broader context better.

>>>Thanks for the insightful comment and suggestion. In the revision, we have combined the Results and Discussion into a single section and extensively revised this part.

We added more background information related to discussion, including the possible causes for the formation of bisexual cones based on extensive morphological observations (**Lines 113-119, 188-191**), a brief introduction of the ABC model for angiosperm flower development (**Lines 322-324**), the out of male/female model for the formation of bisexuality during the origin of flowers (**Lines 324-350**), an explanation of *GGM7* and introduction to the functional studies of the *GGM7* clade genes (**Lines 466-469, 496-498, 507-555**), a short summary of the function and evolutionary history of *SEP* genes (E-class) (**Lines 558-565**), and the role of auxin in flower development (**Lines 925-930**). We also proposed a new hypothesis regarding the emergence of key floral traits, considering the important roles of the *GGM7*, *NLY* and E-class genes and their evolutionary history (**Lines 565-583**), and inferred the potential involvement of the auxin signaling pathway in bisexual cone formation by integrating previous findings with our results (**Lines 930-1002**).

Moreover, we further performed GO enrichment analysis of DEGs in M4 vs. BF4, M4 vs. F4 and F4 vs. BM4 (**Supplementary Fig. 10c, e, f**), and conducted statistical analyses of plant hormones other than auxin (**Supplementary Fig. 10h**). Also, hierarchical clustering analysis of MADS-box genes and the global transcriptome expression profile was carried out using data from all mentioned samples in Niu et al. (2016) (**Supplementary Fig. 2c-d**).

For figures, the key signal region in *in situ* hybridization within the right oval was enlarged (**Fig. 3d-e and Fig. 6c**), and the *in situ* localization result of *PcDAL13* was integrated with the results of dual-LUC transient expression assays of the promoter activity of *PcDAL13* (**Fig. 6b-c**).

Further comments:

Q1: Line 35: 'some important genes': which, how many?

>>> These genes are *PcDAL12*, *PcDAL10*, *PcNEEDLY* and *PcHDG5*. This information has been added in the revision. Please see **line 42**.

Q2: Line 47: I don't think it is true. There exist dioecious angiosperms.

>>>Thanks for the reminder and sorry for the confusion. Yes, there exist dioecious angiosperms, but their male and female organs are secondarily separated (Theißen and Melzer. 2007. *Ann. Bot.*, 100:603-619).

For clarity, we have revised this sentence as follows: “Based on the fact that all land plants but most angiosperms have unisexual reproductive structures, together with the current phylogenetic framework and morphological evidence, bisexuality is considered likely the first step in the origin of flowers followed by compression of the floral axis, although male and female reproductive organs could be secondarily separated like in dioecious angiosperms” (see **lines 85-90**).

Q3: Lines 67-70: This statement needs the appropriate references.

>>>We have cited references as follows:

Theißen, G. et al. How the land plants learned their floral ABCs: the role of MADS-box genes in the evolutionary origin of flower. in *Developmental genetics and plant evolution* (eds. Cronk Q., Bateman R. & Hawkins J.). 173-205 (London: Taylor & Francis, 2002).

Baum, D. A. & Hileman, L. C. A developmental genetic model for the origin of the flower. in *Flowering and its manipulation* (ed. Ainsworth C.). 3-27 (Sheffield, UK: Blackwell Publishing, 2006).

See **line 126**.

Niu, S. et al. A transcriptomics investigation into pine reproductive organ development. *New Phytol.* 209, 1278-1289 (2016).

Feng, X., Yang, X.-M., Yang, Z. & Fan, F.-H. Transcriptome analysis of *Pinus massoniana* Lamb. microstrobili during sexual reversal. *Open Life Sci.* 13, 97-106 (2018).

See **line 113**.

Q4: Line 69: Has it indeed been useful to explain the origin of flowers, how precisely?

>>>The bisexual reproductive structure has long been considered to be unique to angiosperms, and the shift from unisexual to bisexual reproductive units is a crucial step in the origin of flowers (Rudall et al. *Trends Plant Sci.*, 2011, 16: 151-159). According to the euanthial theory, the flower is a uniaxial structure, with carpels and stamens homologous to the macrosporophylls and microsporophylls of gymnosperms (Arber & Parkin, *J. Linn. Soc. Bot.*, 1907, 38:29-80). The morphology of bisexual cone in gymnosperms closely resembles that of an ancestral perianth-less bisexual flower [Theißen et al., 2002, How the land plants learned their floral ABCs: the role of MADS-box genes in the evolutionary origin of flower. in *Developmental genetics and plant evolution* (eds. Cronk Q., Bateman R. & Hawkins J.). 173-205 (London: Taylor & Francis)], exhibiting a transition from unisexual to bisexual, which seems to be consistent with the occurrence of hermaphrodite during the origin of flowers. Consequently, understanding the underlying molecular mechanism may provide clues to the formation of hermaphroditic traits during the origin of flowers.

In order to describe more precisely, in the revision, we replaced “origin of flowers” with “initiation of hermaphrodite during the origin of flowers” (see **line 125**), and added more details as follows: “The euanthial theory supposed that flowers are uniaxial structures, with carpels and stamens homologous to gymnosperm macrosporophylls and microsporophylls, respectively. This resemblance to ancestral perianth-less bisexual flowers is evident in the morphology of bisexual cones in gymnosperms. Consequently, the bisexual cones have long been considered an intermediate state in the origin of flowers, and have been used to explain the initiation

of hermaphrodite during the origin of flowers.” (see **lines 119-126**).

Q5: Line 70: If these cones are aberrant, why they can occur widely? One would assume there is natural selection against aberrant structures.

>>>Cones of most conifers are strictly unisexual (Rudall et al. Trends Plant Sci. 2011, 16: 151-159). All previous studies considered occasionally occurring bisexual cones as aberrations (Rudall et al., Trends Plant Sci. 2011, 16: 151-159), but some recent studies (details see below) showed that bisexual cones could be functional. In the revision, we removed “aberrant” to avoid confusion (see **lines 111, 113**).

Although bisexual cones have been reported in several conifer families such as Pinaceae, Cupressaceae and Araucariaceae, the occurrence of bisexual cones in nature is exceedingly rare. For instance, only 0.5% of *Picea mariana* trees bore 1-6 bisexual strobili, and a low frequency (approximately 1%) of predominantly female monoecious or predominantly male monoecious individuals of *Pinus johannis* changed to monoecious individuals producing bisexual structures (Caron and Powell, Can. J. Bot., 1990, 68: 1826-1830; Flores-Rentería et al., Am. J. Bot., 2013, 100:602-612). Flores-Rentería et al. (2011) proposed that the bisexual structures originated from the common ancestor of gymnosperms and angiosperms and remain conserved. The lack of bisexual structures in gymnosperms may primarily be attributed to natural selection to avoid inbreeding constraints, given the absence of an incompatibility system in gymnosperms. We agree that natural selection acts against bisexual structures in gymnosperms, although additional investigation is needed.

In the revision, we changed this sentence to: “Although exceedingly rare in nature, bisexual cones have been documented in many gymnosperms, particularly conifers like *Agathis*, *Larix*, *Picea*, *Pinus*, *Phyllocladus* and *Saxegothaea*, suggesting that the bisexual structure likely originated from the common ancestor of gymnosperms and angiosperms. A rare occurrence of bisexual structure in gymnosperms may primarily result from natural selection to avoid inbreeding constraints, given the absence of an incompatibility system.” Details see **line 113-119**.

Q6: Line 74-77: I think that the summary of their results could be more explicit.

>>>Thanks for the comment. In the revision, we added the information about the “mostly female” expression profile of bisexual cones, which was the most important conclusion in Niu et al. (2016). This sentence has been rewritten as: “Niu et al. investigated the bisexual cones of *Pinus tabulaeformis*, finding both male and female structures functional. They reported that the transcriptomes of male structures were more similar to female cones, but the expression pattern of MADS-box genes resembled male cones, indicating a “mostly female” gene expression profile in bisexual cones.” Details see **lines 126-130**.

Q7: Line 79-82: My understanding is that male and female structures in bisexual cones were separated immediately after collection (Niu et al. 2016). What the actual

shortcoming here was, was that not early enough developmental stages of bisexual cones were examined in their study.

>>>It has been clarified. This sentence has been changed to: “However, none of these studies conducted separate RNA-seq analysis on the female and male structures of bisexual cones, hindering comparative analysis of gene expression pattern between these structures. Additionally, in the study of Niu et al., the sampling of bisexual cones was not performed early enough to identify direct regulators and further explore the molecular mechanisms initiating bisexual cones.” See **lines 132-159**.

Q8: Line 88: What do you mean by various methylated regions? Can you be more explicit?

>>>Sorry for the mistake, “various” should be “varied”. We have described it in more detail. Please see **lines 165-172**: “Li et al. found that the upregulated genes *AoMSI*, *AoLAP3*, *AoAMS* and *AoLAP5*, with varied methylated CHH regions, might be involved in sexual differentiation in *Asparagus officinalis*. Specifically, during the meiotic stage, *AoMSI* and *AoLAP3* show hypomethylated CHH differentially methylated regions (DMRs) in male flowers, contrasting with female flowers. Additionally, in male flowers at the meiotic stage, *AoAMS* and *AoLAP5* exhibit hypermethylated CHH DMRs, distinguishing them from the premeiotic stage.”

Q9: Line 93: What are those genes named for?

>>>*OGI* and *MeGI* is Japanese for “Male tree” and “Female tree”, respectively (Akagi et al. Science, 2014, 346:646-650). This information has been added in the revision. Please see **line 175**.

Q10: Line 94-96: I think the work by Yakovlev et al. 2020 could be cited here, cf. https://link.springer.com/chapter/10.1007/978-3-030-21001-4_5.

>>>It has been cited. Please see **line 178**.

Q11: Line 100-102; 116-117: Perhaps conclusions cannot be as far reaching at this point as to the origin and evolution of (true) flowers in angiosperms, see also your Lines 62-66 which actually provide evidence.

>>>Thanks for the reminder. In the revision, we use “hermaphrodite or bisexuality” instead of “flowers in angiosperms”. Please see **lines 45, 125**.

Q12: Line 104: Is it a monoecious species? Please add this information here.

>>>Yes, this information has been added in the revision. The sentence has been changed to “In this study, we conducted a comprehensive comparative analysis of transcriptomic, DNA methylomic and hormonal variation in different developmental stages of normal male and female cones and bisexual cones in *Picea crassifolia* (Qinghai spruce), a monoecious species in Pinaceae”. Please see **lines 194-197**.

Q13: Line 105-107: how are these results different from the ones obtained by Niu et al. 2016?

>>>This part has been rewritten. The differences between our study and Niu et al. (2016) are discussed in **lines 258-317** as follows:

“In this study, RNA-seq analysis of unisexual and bisexual cones in *P. crassifolia* revealed highly consistent expression profiles for the biological replicates of male cone (M), female cone (F), male structures of bisexual cones (BM) and female structures of bisexual cones (BF) at each stage (Fig. 2b, c and Supplementary Fig. 1a, b, d, e). The transcriptome expression profiles of BF and BM are similar to those of F and M, respectively (Fig. 2b, c). Differential expression analysis revealed a gradual increase in differentially expressed genes (DEGs) from M2 to M5 relative to M1, indicating changing gene expression patterns during male cone development. In contrast, F3 relative to F1 had more DEGs compared to F2, F4, and F5 relative to F1, possibly due to rapid F3 growth (Supplementary Fig. 1c, f). In addition, among M4, F4, BM4 and BF4, F4 vs. BF4 exhibited the fewest DEGs (Fig. 2d). These results indicate a resemblance between the transcriptome of BF and BM and their respective counterparts, F and M, demonstrating a “bisexual” transcriptome expression pattern in bisexual cones. Niu et al., however, investigated the expression profiles and regulatory mechanisms underlying bisexual cone development of *P. tabuliformis* using RNA-seq and microarray analysis and found a similarity of transcriptome between male structures of bisexual cones and female cones, resulting in a “mostly female” gene expression profile in the bisexual cones. Using the *P. tabuliformis* genome as reference data, we reperformed clustering analysis, including principal component analysis (PCA), Pearson correlation coefficient and hierarchical clustering analysis of transcriptome expression profiles, and hierarchical clustering of MADS-box genes expression profiles. These analyses were conducted on the expression data from all samples reported in Niu et al. (data supported by Shihui Niu). The reanalysis confirmed concordance of the transcriptome and MADS-box gene expression profiles between the male structures of bisexual cones and male cones (Supplementary Fig. 2), as found in *P. crassifolia*. Discrepancies in clustering results from Niu et al. may stem from methodological deviations, given that they used only 3989 DEGs for cluster analysis whereas our reanalysis incorporated over 20,000 genes.”

Q14: Line 105: I think you should introduce to the reader what B-class and C-class genes are and provide examples for those whose precise function is known. There is

not much (concise) information about the evo-devo on reproductive development in seed plants.

>>>We have added introduction of B-class and C-class genes in the revision:
“Since the ABC model established in *Arabidopsis* is also applicable to gymnosperms, where B- and C-class genes govern male cone development, while C-class genes control female cone formation, Theißen et al. proposed the out of male and out of female models based on B-class gene expression changes to explain hermaphrodite formation.” Details please see **lines 322-346**.

Q15: Line 108: please remove: 'Integrated with DNA methylomic analysis' (it is redundant information to what is described next).

>>>Done.

Q16: Line 110-112: It is interesting that you mention *PcNEEDLY* (only expressed in female cones usually) as an example, since this gene has no counterpart in angiosperms.

>>>Thanks for the comments.

NEEDLY, a homologous gene to *LEAFY*, is present in all gymnosperms except *Gnetum* and has been lost in the angiosperm lineage (Frohlich, Nat. Rev. Genet., 2003, 4:559-566). Previous studies proposed that the emergence of bisexual flowers could be associated with the loss of *NEEDLY* (*NLY*) in angiosperms (Frohlich and Parker, Syst. Bot., 2000, 25:155-170).

Our findings indicate that *PcNEEDLY* is expressed in both male and female cones, with a notably higher expression level in female cones (Fig. 3b and Supplementary Fig. 4b), consistent with previous studies (Vazquez-Lobo et al., Evol. Dev., 2007, 9:446-459; Moyroud et al., New Phytol., 2017, 216:469-481). It is worth noting that the variation of DNA methylation in the promoter region of *PcNEEDLY* might affect the expression of this gene (Fig. 6d and Supplementary Fig 8a), thus contribute to the initiation of bisexual cones in *Picea crassifolia*.

We added some discussions about *NEEDLY* genes in the revision as follows:
“Besides that, studies have shown that *NLY* can recognize sequences containing a *LFY* binding motif, inducing flower formation and complementing the *lfy* mutant when expressed in *Arabidopsis thaliana* or *Nicotiana tabacum*. In the initial stage of bisexual cones in *P. crassifolia*, *PcNLY* expression was higher in the female structures compared to normal male and female cones (Fig. 3b and Supplementary Fig. 4b). Consequently, the absence of *GGM7* and *NLY* genes, and retention (or new functionalization) of E-class genes in angiosperms may lead to alterations in the interaction modes among MADS-box genes. These changes could account for the distinct composition of complexes responsible for specifying male and female organ identities between gymnosperms and angiosperms, thus contributing to the formation of key floral traits in angiosperms.” **Lines 565-583**.

Q17: Line 113-114: Auxin has already been implicated in archegonial development (related to the female gametophyte). Some background information on such research could be provided here.

>>>Thanks for the insightful comments. These information have been added in the revision: “Studies on *Arabidopsis* have revealed that disturbances in auxin biosynthesis, transport, or signalling lead to pistil development defects. Specially, auxin activates *AUXIN RESPONSE FACTOR5/MONOPTEROS (ARF5/MP)*, *AUXIN AINTEGUMENTA (ANT)*, and *AINTEGUMENTA-LIKE6/PLETHORA3 (AIL6/PLT3)*, enabling their binding to the *LFY* promoter and directly inducing *LFY* expression. *LFY*, in turn, activates *AP3* and *AG* genes in *Arabidopsis* and potentially regulates B-class genes in gymnosperms. Thus, the auxin-*LFY* module likely plays a crucial role in initiating the bisexual cones in *P. crassifolia*, as evident by the consistent correlation between *LFY* gene expression gradient and IAA concentration gradient (Fig. 3b, 7 Supplementary Fig. 4a, 10h)” (See **lines 925-1002**).

Q18: Line 116-117: how does this study provide new clues on hermaphroditic flowers?

>>>We have rewritten this part as follows: “We aim to: (1) evaluate which hypothesis of origin of bisexuality is more consistent with the transcriptome and MADS-box gene expression patterns of bisexual cones; (2) reveal the role of DNA methylation in bisexual cone initiation and its influence on cone development-related genes; and (3) investigate hormonal influences on the development of bisexual cones and identify specific hormonal changes that promote its development”. Please see **lines 197-203**.

Q19: Lines 67, 70, 384: The term "abnormal" or aberrant in the reproductive development of gymnosperms has never been explained here, in the sense of how often these structures occur and what is the reason for their occurrence. This is perhaps the most intriguing question.

>>>Thanks for the comments. The terms "abnormal" or "aberrant" have been removed in the revision, as explained in the response to **Q5** above.

In fact, the occurrence of bisexual cones in nature is exceedingly rare. For instance, only 0.5% of *Picea mariana* trees bore 1-6 bisexual strobili, and a low frequency (approximately 1%) of predominantly female monoecious or predominantly male monoecious individuals of *Pinus johannis* changed to monoecious individuals producing bisexual structures (Caron and Powell, *Can. J. Bot.*, 1990, 68: 1826-1830; Flores-Rentería et al., *Am. J. Bot.*, 2013, 100:602-612). As a consequence, Flores-Rentería et al. (2011) proposed that the bisexual structures originated from the common ancestor of gymnosperms and angiosperms and remain conserved. That is, gymnosperms possess the capacity to produce bisexual structures similar to those of angiosperms, without the need for environmental perturbations to induce them. This phenomenon is supported by the presence of homologous genes regulating sex

expression in gymnosperms, as found in angiosperms (Theißen et al., Development, 2016, 143:3259-3271). Therefore, they deduced that a rare occurrence of bisexual structures in gymnosperms may primarily be attributed to natural selection to avoid inbreeding constraints, given the absence of an incompatibility system in gymnosperms. In addition, due to that sexual inconstancy was only detected in some unisexual individuals but not in others, Flores-Rentería et al. (2013) suggested that genetic plasticity might contribute to the formation of complex sexual systems of *Pinus johannis*. Therefore, in this study, we conducted a comprehensive comparative analysis of transcriptomic, DNA methylomic and hormonal variation in different developmental stages of normal male and female cones and bisexual cones in *Picea crassifolia*, a monoecious species in Pinaceae.

In the revision, we have added more background information as follows: “Although exceedingly rare in nature, bisexual cones have been documented in many gymnosperms, particularly conifers like *Agathis*, *Larix*, *Picea*, *Pinus*, *Phyllocladus* and *Saxegothaea*, suggesting that the bisexual structure likely originated from the common ancestor of gymnosperms and angiosperms. A rare occurrence of bisexual structure in gymnosperms may primarily result from natural selection to avoid inbreeding constraints, given the absence of an incompatibility system” Please see **lines 113-119**.

“Previous studies have shown that global methylation levels of CG and CHG in gymnosperms are much higher than those in angiosperms, and Flores-Rentería et al. suggested that genetic plasticity might contribute to the formation of complex sexual systems of *Pinus johannis* because sexual inconstancy was only detected in some unisexual individuals but not in others.” Please see **lines 178-191**.

Q20: Line 124-126: to my knowledge NLY has no counterpart in angiosperms thus cannot help to explain there the appearance of hermaphroditic (bisexual) flowers. Or is this not what was meant here to describe?

>>>Frohlich and Parker (Syst. Bot., 2000, 25:155-170) proposed the mostly male theory based on their analysis of the evolutionary history of the *LFY* gene family in seed plants. They found that angiosperms possess a single copy of the *LFY* gene, whereas gymnosperms have two copies, *LFY* and *NLY*, with *NLY* potentially being lost before the origin of angiosperms. Additionally, studies of Mellerowicz et al. (Planta, 1998, 206:619-629) and Mouradov et al. (Proc. Nat. Acad. Sci. USA, 1998, 95:6537-6542) in *Pinus radiata* revealed that *LFY* gene expression is predominantly in male cones, whereas *NLY* gene expression primarily in female cones. Based on these evidences, Frohlich and Parker (2000) hypothesized that in the common ancestor of extant angiosperms and gymnosperms, *LFY* and *NLY* genes determined male and female reproductive structure, respectively. They suggested that the *LFY* gene, expressed in angiosperm flowers, evolved from the gene responsible for male reproductive structures in gymnosperms. The loss of the female-determining gene *NLY* in angiosperms resulted in the loss of many downstream genes regulated by *NLY*. Consequently, they concluded that the majority of genes expressed in angiosperm bisexual flowers are homologous to those expressed in male cones of gymnosperms,

supporting the mostly male theory (Forhlich and Parker, Syst. Bot., 2000, 25:155-170). Therefore, the loss of *NLY* helps explain the emergence of bisexuality.

In the revision, we added some discussions about the *NLY* gene as follows: “Besides that, studies have shown that *NLY* can recognize sequences containing a *LFY* binding motif, inducing flower formation and complementing the *lfy* mutant when expressed in *Arabidopsis thaliana* or *Nicotiana tabacum*. In the initial stage of bisexual cones in *P. crassifolia*, *PcNLY* expression was higher in the female structures compared to normal male and female cones (Fig. 3b and Supplementary Fig. 4b). Consequently, the absence of *GGM7* and *NLY* genes, and retention (or new functionalization) of E-class genes in angiosperms may lead to alterations in the interaction modes among MADS-box genes. These changes could account for the distinct composition of complexes responsible for specifying male and female organ identities between gymnosperms and angiosperms, thus contributing to the formation of key floral traits in angiosperms” (see **lines 565-583**).

Q21: Line 377-379: this is definitely not part of a Results section!

>>>Sorry for the confusion. We have combined the Results and Discussion into a single section in the revision. **Line 205**.

Q22: Line 382-390: This is redundant to previous information provided.

>>> Thank you for your reminder. We deleted the first two sentences in the revision. Please see **line 1048**.

Q23: Line 384-385: In which sense are those bisexual cones resembling primitive flowers? Something about homology and analogy of reproductive structures should have been introduced and discussed.

>>>Thanks for the good suggestion. Based on the current phylogenetic framework, the euanthial theory is gradually widely accepted (Shan & Kong, Chin. Sci. Bull., 2017, 62:2323-2334). The euanthial theory supposed that a flower is uniaxial structure, with carpels and stamens homologous to gymnosperm macrosporophylls and microsporophylls, respectively (Arber & Parkin, J. Linn. Soc. Bot., 1907, 38:29-80). Consequently, the morphology of bisexual cones in gymnosperms closely resembles that of ancestral perianth-less bisexual flowers [Theißen et al., 2002, How the land plants learned their floral ABCs: the role of MADS-box genes in the evolutionary origin of flower. in Developmental genetics and plant evolution (eds. Cronk Q., Bateman R. & Hawkins J.). 173-205 (London: Taylor & Francis)] and have been utilized to elucidate the initiation of hermaphroditism during the origin of flowers.

We have added this background in the revision as follows: “The euanthial theory supposed that flowers are uniaxial structures, with carpels and stamens homologous to

gymnosperm macrosporophylls and microsporophylls respectively. This resemblance to ancestral perianth-less bisexual flowers is evident in the morphology of bisexual cones in gymnosperms. Consequently, the bisexual cones have long been considered an intermediate state in the origin of flowers, and have been used to explain the initiation of hermaphrodite during the origin of flowers.” Please see **lines 119-126**.

Q24: Line 390-397: How are those results different from Niu et al., 2016?

>>> The differences between our study and Niu et al. (2016) are discussed in **lines 258-317**. Please see **Q13**.

Q25: Line 398: Please explain somewhere in the manuscript the exact functional mode of action of GGM7 genes and provide the explanation of the abbreviation for the GGM7 genes. Also, it is not clear from your writing what is the new knowledge from your study and what is already known (a reference in your discussion).

Line 405-408: A lot more information on GGM7 genes' function is needed in your discussion to more fully grasp their exact importance in flower evolution.

>>>Thanks for the insightful comments. In the revision, we provided the explanation of the abbreviation for the *GGM7* genes, and added more details for the exact functional mode of action of *GGM7* genes as follows:

“*G. gnemon MADS7 (GGM7)* was firstly cloned and sequenced from *Gnetum gnemon*, with possible corresponding orthologous genes found in ferns and bryophytes. However, it lacks a counterpart in angiosperms. In this study, we identified two *GGM7* genes, *PcDAL21* (c98512_g1_i1) and *PcDAL10* (c118861_g1_i1), homologous to *PaDAL21* and *PaDAL10* of *Picea abies*, respectively. *PcDAL21* exhibited negligible expression in male cones but showed substantial expression in early-stage female structures, with higher expression levels observed in BF4 than in BM4, presenting an expression pattern contrasting with that of B-class genes (Fig. 3b, c). This expression pattern was confirmed by *in situ* localization, where specific *PcDAL21* signals were localized predominantly at the base of the female structure within the bisexual cone, while being relatively weak in the microsporophyll (Fig. 3e). The specific expression in female cones of *DAL21* has also been observed in *Picea abies* and *Cunninghamia lanceolata*, indicating a correlation between *DAL21* expression initiation and the onset of ovuliferous scale primordia. Considering B-class genes' high expression at the base and low levels at the top of the reproductive axis, we infer that *PcDAL21* may function in female cones similar to how B-class genes act on male cones, establishing female identity and potentially antagonizing B-class genes. However, further experiments are needed to confirm *DAL21*'s role in forming floral quartet-like complexes in female cones of *P. crassifolia* or even gymnosperms. The expression of *PcDAL10*, another copy of *GGM7*, was higher in bisexual cones than in male and female cones at corresponding developmental stages (Fig. 3b and Supplementary Fig. 4c). Previous studies revealed specific *PaDAL10*'s expression in reproductive structures of *P. abies*. Transgenic

Arabidopsis plants expressing *PaDAL10* exhibited notable morphological changes in sepals, petals and stamens, suggesting its interaction with B- and C-class genes. This hypothesis was validated through yeast two-hybrid assays involving *PtDAL10*, an orthologue of *PaDAL10* in *Pinus tabuliformis*, which exhibited widespread interactions with other MADS-box genes, including B-class, C-class, *SEP/AGL6*, and others. Therefore, both ectopic expression of *PcDAL21* and increased expression of *PcDAL10* might be essential for the initiation of bisexual cones.” Please see **lines 466-557**.

Q26: Line 406: No previous introduction about E-class genes was provided.

>>>Thanks for the suggestion. It has been added in the revision, see **lines 558-583**: “Additionally, *SEP* genes (E-class), which act as mediators in the formation of male- and female-specifying complexes in angiosperms, have not been identified outside the angiosperm lineage. While some gymnosperm genes are phylogenetically close to the *AGL6* subfamily, a clade closely related to E-class genes, it remains a matter of debate whether these genes are true orthologs of angiosperm *AGL6* clade or if they are instead sister to the *AGL6/SEP* clade. Notably, a study has shown that in *Gnetum gnemon*, B- and C-class proteins can directly interact without the need for *SEP* or *AGL6* genes as mediators. Besides that, studies have shown that *NLY* can recognize sequences containing a *LFY* binding motif, inducing flower formation and complementing the *lfy* mutant when expressed in *Arabidopsis thaliana* or *Nicotiana tabacum*. In the initial stage of bisexual cones in *P. crassifolia*, *PcNLY* expression was higher in the female structures compared to normal male and female cones (Fig. 3b and Supplementary Fig. 4b). Consequently, the absence of *GGM7* and *NLY* genes, and retention (or new functionalization) of E-class genes in angiosperms may lead to alterations in the interaction modes among MADS-box genes. These changes could account for the distinct composition of complexes responsible for specifying male and female organ identities between gymnosperms and angiosperms, thus contributing to the formation of key floral traits in angiosperms.”

Q27: Line 410: Can you provide more information about the exact methylation reconfiguration that was observed and its implication? And without having to consult figures and supplement material?

Line 411-414: ditto

>>> We added the implication of DNA reconfiguration in **lines 627-630** as follows:

“These findings suggest CG and CHG DNA methylation reconfiguration in bisexual cones, indicating genome-wide changes of DNA methylation patterns without the massive disappearance of DNA methylation”

There are three evidences for CG and CHG DNA methylation reconfiguration in the bisexual cones: (1) The global CG and CHG methylation levels in bisexual cones differ from those in normal cones (Fig. 4a and Supplementary Fig. 6a); (2) In bisexual cones, methylated CG and CHG sites exhibit concentrated patterns within specific

unigenes in female structures, contrasting with the opposite trends observed in male structures, in comparison to normal cones (Fig. 4c-d and Supplementary Fig. 6c); (3) The number and DNA methylation differences of DMCs differed significantly between bisexual cones and normal cones, signifying substantial changes in methylations at multiple sites (Fig. 5a-b). The detailed information has been provided in **lines 595-630**:

“In general, compared to the other three tissues, BF4 exhibited the highest global CG and CHG methylation levels and the lowest percentage of CG and CHG body-methylated genes. Nonetheless, there was no significant difference in the number of methylated sites across four tissues (Fig. 4a-c and Supplementary Fig. 6a-c). Notably, the percentages of CG and CHG body-methylated genes were highest in BM4 (Fig. 4c and Supplementary Fig. 6c). Through the analysis of DNA methylation site density in body-methylated genes, we found that genes with high CG and CHG methylation site density exhibited the highest occurrence in BF4 and the lowest in BM4 (Fig. 4d). These results indicate altered global methylation pattern in bisexual cones compared to normal male and female cones. BF4 showed slightly increased global CG and CHG methylation levels, with concentrated methylated CG and CHG sites in specific unigenes, while BM4 exhibited the opposite trends, suggesting distinct methylation strategies between the two. In addition, we observed a high overlap of body-methylated genes across various tissues, with BF4-specific body-methylated genes being the least. Significant differences in CG and CHG methylation levels were noted in body-methylated genes in M4 vs. F4 and F4 vs. BF4 (Fig. 4e, f and Supplementary Data 2-5). Furthermore, there were more differentially methylated cytosines (DMCs) and DMRs in M4 vs. BM4 and F4 vs. BF4, with greater DNA methylation level differences of DMCs in M4 vs. BM4 and F4 vs. BF4 compared to M4 vs. F4 and BM4 vs. BF4 (Fig. 5a, b). These findings suggest CG and CHG DNA methylation reconfiguration in bisexual cones, indicating genome-wide changes of DNA methylation patterns without the massive disappearance of DNA methylation.”

Q28: Line 415: In my opinion, the function of localized auxin implication should be more emphasized for this research.

>>>This suggestion has been followed. First, in the section “Auxin could enhance the femaleness of *Picea crassifolia*”, we incorporated background information regarding Auxin’s function in archegonial development and inferred potential roles of the auxin signaling pathway in bisexual cone information, drawing on prior research and our finds. See **lines 925-1002**: “Studies on *Arabidopsis* have revealed that disturbances in auxin biosynthesis, transport, or signalling lead to pistil development defects. Specially, auxin activates *AUXIN RESPONSE FACTOR5/MONOPTEROS (ARF5/MP)*, *AUXIN AINTEGUMENTA (ANT)*, and *AINTEGUMENTA-LIKE6/PLETHORA3 (AIL6/PLT3)*, enabling their binding to the *LFY* promoter and directly inducing *LFY* expression. *LFY*, in turn, activates *AP3* and *AG* genes in *Arabidopsis* and potentially regulates B-class genes in gymnosperms. Thus, the auxin-*LFY* module likely plays a crucial role in initiating the bisexual cones in *P.*

crassifolia, as evident by the consistent correlation between *LFY* gene expression gradient and IAA concentration gradient.”

Second, we added the GO enrichment analysis of DEGs in M4 vs. BF4, BM4 vs. F4 and M4 vs. F4 (Supplementary Fig. 10c, e, f) and observed a noteworthy presence of GO terms associated with auxin in up-regulated genes of F4 vs. BF4, M4 vs. BF4, BM4 vs. BF4 and BM4 vs. F4, and in down-regulated genes of M4 vs. BM4 and M4 vs. F4. See **lines 914-918**: “Interestingly, genes related to auxin signal transduction, transport and response were highly expressed in BF compared to BM during bisexual cone initiation. Auxin-related pathways were enriched in upregulated genes in F4 vs. BF4, M4 vs. BF4 and BM4 vs. F4, and in downregulated genes in M4 vs. BM4 and M4 vs. F4 (Supplementary Fig. 10a-g).”

Third, we added statistical analysis of other important hormones in the revision and found that except IAA, there are no significant differences in the content of other hormones among BM, BF, F and M during the initiation of bisexual cones (Supplementary Fig. 10h). Please see **lines 1006-1009**: “However, except IAA, there were no significant differences in the content of other hormones among BM, BF, F and M during cone development (Supplementary Fig. 10h and Supplementary Data 10).”

Finally, we added more information in the section “Conclusion”. Please see **lines 1098-1102**: “Finally, we observed an auxin concentration gradient within bisexual cones, accompanied by heightened expression of auxin-related genes in the female reproductive structures of bisexual cones, suggesting a potential significant role for auxin in bisexual cone initiation.”

Q29: Line 417-420: The exploration of the mechanism of bisexual cone initiation is unclear unless a mechanistic model can be developed on the study's results. And this should be better highlighted here.

>>>Following the suggestion, we explained the potential mechanisms of bisexual cone initiation in the revision based on our multi-omics study. Please see **Figure 7** and **lines 1103-1140**:

“Based on our multi-omics study, we propose a model explaining the molecular mechanism underlying bisexual cone initiation. We hypothesize that some individuals of Qinghai spruce may undergo DNA methylation reconfiguration and alterations in auxin concentration during male cone development. These changes facilitate shifts in *LFY* and *NLY* expression, leading to the establishment of an expression gradient along reproductive axis, with the highest expression at the apex. Upon reaching specific expression thresholds for *LFY* and *NLY*, the initiation of *DAL21* expression occurs. Due to the functional antagonism between *DAL21* and B-class genes, coupled with the regulatory influence of DNA methylation, the expression of B-class genes becomes progressively weaker and ultimately inactive. Consequently, this results in the changes of sex-determining protein complexes at the tip of the reproductive axis, thereby promoting megasporophyll development. In this intricate process, *DAL10*, whose expression is upregulated in bisexual cones through methylation or regulation

of *LFY* and *NLY* genes, plays a pivotal role in shaping the protein interaction network during sex determination (Fig. 7).”

Q30: Figures 1a-c and 2a, 3b, 3e, 3f and 7: It would be important to indicate the developmental stage of those cones, and in all cases. All those codes are not explained in the figure legends.

>>>It has been followed. We have provided the exact collection times for these cones in the figure legends. Please see **lines 1756, 1761-1763, 1769-1770, 1772-1773, 1802, 1819-1820.**

Q31: Figure 3e,f: While an *in situ* hybridisation experiment is worthwhile, it is very difficult to properly discern the localization in those images.

>>>Sorry for the negligence. The key signal regions in *in situ* hybridization have been enlarged. Please see **Fig. 3d-e** and **Fig. 6c.**

Q32: Figure 4a,b, f(right plot): Have any statistical tests been performed on those patterns?

>>>The methylation data in Figure 4a and 4b were obtained by combining data from three biological replicates, making it unfeasible to conduct significance tests. In fact, we individually assessed methylation levels and the number of methylated sites for each duplicate and performed significance tests, as detailed in Supplementary Figure 6. In addition, the CHH methylation levels of body-methylated genes among M4, F4, BM4 and BF4 lack significant differences, as denoted in Figure 4f. The pertinent information is detailed in the figure legend.

Q33: Figure 5a: Are those patterns significantly different?

>>> Figure 5a presents a bar chart illustrating quantity statistics, specifically the number of DMCs and DMRs among M4, F4, BM4 and BF4. Therefore, the test for assessing the significance of difference was not applicable here. To make the meaning conveyed by the bar chart clearer, we added labels containing the raw statistics on the bars. A detailed description of the bar chart can be found in **lines 624-627:**

“Furthermore, there were more differentially methylated cytosines (DMCs) and DMRs in M4 vs. BM4 and F4 vs. BF4, with greater DNA methylation level differences of DMCs in M4 vs. BM4 and F4 vs. BF4 compared to M4 vs. F4 and BM4 vs. BF4.”

Q34: Figures 6, 7: contain a lot of information. But I think the overall outline could be improved. For example, the reporter/effector study of the DAL13 promoter could be

merged with the *in situ* localization results (which should become more evident), and the legend for Figure 7 needs to be improved. I didn't find the description very clear.

>>> Following the suggestion, we added the *in situ* localization result of *PcDAL13* in Figure 6. Additionally, we renewed the legend of Figure 7. Detailed please see **lines 1811-1820**.

“The graphical representation utilizes horizontal lines with solid and hollow dots to represent DNA methylation reconfiguration, while the orange circles labelled with IAA depict the concentration gradient of IAA. Upregulated genes are highlighted in red, while downregulated genes in blue. Promoter hypomethylation is denoted by “m” on a white background, whereas gene-body hypermethylation is indicated on a red background. Positive regulatory relationships are illustrated by orange arrows. The expression gradient of *DAL21*, *LFY* and *NLY* in bisexual cones is represented by a gradual blue triangle, while the gradual light purple triangle shows the expression gradient of B-class genes in bisexual cones. The male cone was taken on 1 April, 2019, and the bisexual cone was photoed on 10 April, 2019.”

In addition, we summarized our findings and propose a new model explaining the molecular mechanism of bisexual cone initiation based on Fig. 7, which is described in the Conclusion section. Please see **lines 1103-1140**.

Reviewer #3 (Remarks to the Author):

This manuscript performed transcriptomic, DNA methylomic, and metabolomic analyses to explore the molecular mechanism of the bisexual cone initiation in *Picea crassifolia*. The authors concluded that the developmental mechanism of bisexual cones is consistent with the out of male model and MADS-box family genes and their regulated genes are essential for the initiation of the female reproductive structure of bisexual cones. They also found that the expression patterns of some cone development related genes might be affected by the DNA methylation variation and the production of the female reproductive structure of bisexual cones might be closely related to the auxin content variation in the initiation of bisexual cones. The study makes a contribution to the molecular mechanism of the bisexual cone initiation in conifers and the evolution of flowers of angiosperms.

>>>Thanks for the positive comments.

Some general comments conducted:

1. Expression profiles of the male structure of bisexual cones are more similar to that of male cones in both *P. crassifolia* and *Pinus tabuliformis*, which is not consistent with the results mentioned in Niu et al (2016) (Figure 2b,c; Supplementary Figure 2; P7 L146-147; P7 L159- P8 L161). The authors claimed that the reason for this inconsistency might be due to the deviation of analysis methods (P8 L161-162). In

addition to Pearson correlation coefficients and principal component analysis (PCA) performed in this study, more convincing analysis methods are needed to confirm the results.

>>>Following the suggestion, we conducted hierarchical clustering analysis on transcriptome expression profiles and MADS-box genes expression levels extracted using genome annotation of *Pinus tabuliformis*. These findings, in accordance with both PCA and Pearson correlation coefficient analyses, confirm the identical gene expression pattern between the male structure of bisexual cones and male cones. Please see **Supplementary Figure 2c-d**.

Additionally, in contrast to Niu et al. (2016), who employed a *de novo* assembled transcriptome of *Pinus tabuliformis* as a reference, and used 3989 differently expressed genes in M vs. MT for cluster analysis, our study used the *Pinus tabuliformis* genome as a reference, and analyzed more than 20,000 genes with expression levels (TPM value) exceeding 10 in at least one individual for PCA and Pearson correlation coefficient analysis. This likely explains the discrepancies between our PCA and Pearson correlation analysis results and those reported by Niu et al. (2016).

This information has been added in the revision as follows: “Discrepancies in clustering results from Niu et al.’s study may stem from methodological deviations, given that they used only 3989 DEGs for cluster analysis whereas our reanalysis incorporated over 20,000 genes” (see **lines 315-317**).

2. The authors found that F4 vs. BF4 has the least number of differentially expressed genes among M4, F4, BM4, and BF4 (Figure 2d; P7 L148-150). It appears that female cones of bisexual cones are more likely to arise from female cones of unisexual cones than from male cones of unisexual cones. However, the authors stated that the out of male model hypothesis could be used to explain the origin of bisexual cones of *P. crassifolia* after comparing the expression patterns of MIKCC-type MADS-box genes, LFAFY, and NEEDLY and performing six years of field observation (Figure 3; Supplementary Figures 3, 4). A comment on this conflict is needed.

>>>Thanks for the comments. According to Theißen et al. [2002, How the land plants learned their floral ABCs: the role of MADS-box genes in the evolutionary origin of flower. in *Developmental genetics and plant evolution* (eds. Cronk Q., Bateman R. & Hawkins J.). 173-205 (London: Taylor & Francis)], who proposed the out of male and out of female models based on B-class gene expression changes, the former suggested that reduced B-class gene expression in the upper part of the male cone leads to ectopic ovule development, while the latter assumed that ectopic expression of B-class genes in the basal part of the female cone results in the ectopic development of male reproductive units.

In our study, F4 and BF4 exhibited the lowest DEG numbers, indicating a significant female transcriptome expression profile for BF (Fig. 2d). In fact, our observations found that the male structure of bisexual cones develops first, with the female structure

growing at the tip of the male structure, and nearly all bisexual cones exhibit growth in the male cone position (Fig. 1a), consistent with the findings of Caron and Powell (1990) and Flores-Rentería et al. (2011) (Caron and Powell, *Can. J. Bot.*, 1990, 68: 1826-1830; Flores-Rentería et al., *Am. J. Bot.*, 2011. 98:130-139). These results suggest a sex change occurring at the tip of male cone, leading to the formation of bisexual cones. Furthermore, our data reveal that the transcriptome expression profiles and MADS-box gene expression patterns of BM and BF resemble those of M and F, respectively (Fig. 2b, c, d and Fig. 3b). Notably, B-class gene expression is reduced in the female structure of bisexual cones. Consequently, our results support the out of male model.

We have provided the background about the out of male and out of female models in **lines 322-350**: “Since the ABC model established in *Arabidopsis* is also applicable to gymnosperms, where B- and C-class genes govern male cone development, while C-class genes control female cone formation, Theißen et al. proposed the out of male and out of female models based on B-class gene expression changes to explain hermaphrodite formation. The former suggested that reduced B-class gene expression in the upper part of male cone led to ectopic ovule development, while the latter assumed that ectopic expression of B-class genes at the base of female cone resulted in the ectopic development of male reproductive units.”

In addition, we have revised the text in **lines 357-361**:

“In addition, observations over six years showed that almost all bisexual cones exhibited growth in the male cone position (Fig. 1a), consistent with the findings of Caron and Powell and Flores-Renteria et al. These observations, including transcriptome expression profiles, and B-class gene expression patterns, support the out of male model.”

3. PcHDG5 can bind to the promoters of PcDAL12, PcDAL13, and c113171_g1_i1 (Figure 6b; Supplementary Figure 7b,c; P13 L284-288). The authors concluded that PcHDG5 might activate transcription of downstream MADS-box genes by binding to the core-binding motif (P13 L288-290). Do the authors know which motifs PcHDG5 directly binds to?

>>> Following the suggestion, we found binding sites for *AtHDG5* with high confidence in the promoter region of *PcDAL13*, as confirmed by the JASPAR database (9th version, <http://jaspar.genereg.net/>) (**Supplementary Table 3**).

Phylogenetic analysis revealed a close relationship between *PcHDG5* and the *Arabidopsis thaliana* genes *AtHDG5* and *AtHDG4* (Supplementary Figure 7a). Consequently, we inferred that the *PcHDG5* may bind to the promoter region of *PcDAL13* through shared conserved motifs. Supplementary Table 3 provides the detailed results of predicted *AtHDG5* (*Arabidopsis thaliana*) binding sites located within the promoter of *PcDAL13* gene according to the JASPAR database. In addition, this information has been added in the revision. See **lines 732-734**:

“Additionally, binding sites for *AtHDG5* were found with high confidence in the promoter region of *PcDAL13*, as confirmed by the JASPAR database (9th version, <http://jaspar.genereg.net/>) (Supplementary Table 3).”

4. Both the auxin content and the expression of auxin signal related genes differ among the different stages of unisexual cones and bisexual cones (Supplementary Figure 10). How does the auxin signal pathway couple with MADS-box gene related regulation pathway at the initiation stage of bisexual cones? Also, it is better to add the GO enrichment analysis of DEGs in M4 vs. BF4.

>>> Thanks for the good suggestion. Following the suggestion, we performed GO enrichment analysis on DEGs in M4 vs. BF4, M4 vs F4 and BM4 vs. F4 (Supplementary Figure 10c, e, f). The results revealed enrichment of GO terms related to auxin biosynthetic process, response and transport in upregulated genes of F4 vs. BF4, M4 vs. BF4, BM4 vs. BF4, BM4 vs F4, as well as in downregulated genes of M4 vs. BM4 and M4 vs. F4. These findings suggest that auxin plays a significant role during the initiation of bisexual cones in *Picea crassifolia*.

This information has been added in the revision. Please see **lines 914-918**: “Interestingly, genes related to auxin signal transduction, transport and response were highly expressed in BF compared to BM during bisexual cone initiation. Auxin-related pathways were enriched in upregulated genes in F4 vs. BF4, M4 vs. BF4 and BM4 vs. F4, and in downregulated genes in M4 vs. BM4 and M4 vs. F4 (Supplementary Fig. 10a-g).”

Auxin can activate *AUXIN RESPONSE FACTOR5/MONOPTEROS (ARF5/MP)*, *AUXIN AINTEGUMENTA (ANT)*, and *AINTEGUMENTA-LIKE6/PLETHORA3 (AIL6/PLT3)*, which can bind to the *LFY* promoter, directly inducing *LFY* expression (Yamaguchi et al., *Plant physiol.*, 2016, 170:283-293; Yamaguchi et al., *Dev. Cell*, 2013,24:271-282). *LFY*, in turn, acts as a direct activator of both *AP3* and *AG* genes in *Arabidopsis* and potentially serves as an upstream regulator of B-class genes in gymnosperms (Lamb et al., *Development*, 2001, 129:2079-2086; Lohmann et al., *Cell*, 2001, 105:793-803; Moyroud et al., *New phytol.* 2017, 216:469-481). Thus, the auxin-*LFY* module likely plays a crucial role in the initiation of the bisexual cone in *Picea crassifolia*, as suggested by the consistent correlation between the expression gradient of the *LFY* gene and the concentration gradient of IAA (Fig. 3b, Supplementary Fig. 4a and Supplementary Fig. 10h).

We have added this information in **lines 925-1002**: “Studies on *Arabidopsis* have revealed that disturbances in auxin biosynthesis, transport, or signalling lead to pistil development defects. Specially, auxin activates *AUXIN RESPONSE FACTOR5/MONOPTEROS (ARF5/MP)*, *AUXIN AINTEGUMENTA (ANT)*, and *AINTEGUMENTA-LIKE6/PLETHORA3 (AIL6/PLT3)*, enabling their binding to the *LFY* promoter and directly inducing *LFY* expression. *LFY*, in turn, activates *AP3* and *AG* genes in *Arabidopsis* and potentially regulates B-class genes in gymnosperms. Thus, the auxin-*LFY* module likely plays a crucial role in initiating the bisexual cones in *P. crassifolia*, as evident by the consistent correlation between *LFY* gene expression gradient and IAA concentration gradient (Fig. 3b, 7 Supplementary Fig. 4a, 10h).”

5. Figures 1c, 2a, 4b and Supplementary Figure 1c,f are not referred in the main text.

>>>Thanks for the reminder, these figures have been cited in the revision. Figure 2a: **line 1163**; Fig. 1c: **line 1161**; Fig. 4b: **line 599**; Supplementary Fig. 1c,f: **line 267**.

6. The abbreviations should be stated in the figure legend of Supplementary Figure 2.

>>>It has been added in the revision.

“**a-c** Principal component analysis (PCA), Pearson correlation coefficient and hierarchical clustering analysis of transcriptome expression profiles.”

7. It seems that Figure 4a, b, and c are the same with Supplementary Figure 6.

>>>Sorry for the confusion. The methylation data in Figure 4a, b and c were obtained by combining data from three biological replicates. Supplementary Figure 6 shows the individually assessed methylation levels and the number of methylated sites for each replicate. To avoid confusion, we have specified the data sources in the figure legends of Fig. 4 and Supplementary Fig. 6.

In addition, five figures were revised as follows:

Figure 3: The bar graph for qRT-PCR analysis was replaced with a box plot (Fig. 3c) and the key signal regions in *in situ* hybridization were enlarged within the right oval (Fig. 3d-e).

Figure 4: Statistically significant differences from the two-tailed Student's *t*-test were indicated in fig. 4f.

Figure 5: In Fig. 5a, labels containing raw statistics were added on the bars; in Fig. 5b, results of the two-tailed Wilcoxon test were denoted with asterisks.

Figure 6: The *in situ* localization result of *PcDAL13* was incorporated into Fig. 6c.

Figure 7: Our findings were summarized, and a new model explaining the molecular mechanism of bisexual cone initiation based on Fig. 7 was described in the Conclusion section.

REVIEWERS' COMMENTS:

Reviewer #1 (Remarks to the Author):

All of my concerns have been addressed.

Reviewer #2 (Remarks to the Author):

The authors have satisfactorily responded to my previous concerns. However, the manuscript requires some careful English revision before being published.

Reviewer #3 (Remarks to the Author):

The authors have addressed my previous comments and concerns in their revision. I recommend this article to be accepted for publication.

Response to reviews

Reviewer #1 (Remarks to the Author):

All of my concerns have been addressed.

>>> Thanks for your insightful comments again, which have markedly enhanced the quality of our manuscript.

Reviewer #2 (Remarks to the Author):

The authors have satisfactorily responded to my previous concerns. However, the manuscript requires some careful English revision before being published.

>>> Your insightful comments have significantly enhanced the quality of our work. In the revision, the English language and grammar have been carefully refined throughout the manuscript.

Reviewer #3 (Remarks to the Author):

The authors have addressed my previous comments and concerns in their revision. I recommend this article to be accepted for publication.

>>> Thank you for your comprehensive manuscript review and insightful comments again.